# O-linked N-acetylglucosamine transferase is involved in fine regulation of flowering time in winter wheat

Min Fan[1,2], Fang Miao[1,3], Haiyan Jia[1,2], Genqiao Li[1,4], Carol Powers [1], Ragupathi Nagarajan [1], Phillip D. Alderman [1], Brett F. Carver[1], Zhengqiang Ma [2] & Liuling Yan [1✉]

Vernalization genes underlying dramatic differences in flowering time between spring wheat and winter wheat have been studied extensively, but little is known about genes that regulate subtler differences in flowering time among winter wheat cultivars, which account for approximately 75% of wheat grown worldwide. Here, we identify a gene encoding an O-linked N-acetylglucosamine (O-GlcNAc) transferase (OGT) that differentiates heading date between winter wheat cultivars Duster and Billings. We clone this TaOGT1 gene from a quantitative trait locus (QTL) for heading date in a mapping population derived from these two bread wheat cultivars and analyzed in various environments. Transgenic complementation analysis shows that constitutive overexpression of TaOGT1b from Billings accelerates the heading of transgenic Duster plants. TaOGT1 is able to transfer an O-GlcNAc group to wheat protein TaGRP2. Our findings establish important roles for TaOGT1 in winter wheat in adaptation to global warming in the future climate scenarios.

[1] Department of Plant and Soil Sciences, Oklahoma State University, Stillwater, OK, USA. [2] Crop Genomics and Bioinformatics Center and National Key Laboratory of Crop Genetics and Germplasm Enhancement, Nanjing Agricultural University, Jiangsu, Nanjing, PR China. [3]Present address: College of Life Science, Northwest A & F University, Yangling, Shaanxi, PR China. [4]Present address: Wheat, Peanut and Other Field Crops Research Unit, USDA–ARS, Stillwater, OK, USA. ✉email: liuling.yan@okstate.edu

Wheat (*Triticum aestivum*, $2n = 6\times = 42$, AABBDD) is the most widely grown crop worldwide and has become a staple crop of agriculture and human nutrition globally[1]. Flowering time or heading date is a critical trait of wheat that underlies its adaptation to diverse climatic environments and cropping seasons. Wheat is classified into two distinct types: winter wheat, which requires exposure of the seedlings to low temperatures to accelerate flowering (vernalization), and spring wheat, which lacks this vernalization requirement[2,3]. Winter wheat cultivars account for ~75% of the wheat grown worldwide and will become more important as demand for food increases in future as a result of the accelerating population growth.

The four wheat vernalization genes *VRN1*, *VRN2*, *VRN3*, and *VRN4* have been cloned using a positional cloning approach. *VRN1* is a promoter of flowering that is upregulated by vernalization[4] and encodes a MADS-box transcription factor with high similarity to the *Arabidopsis thaliana* (Arabidopsis) meristem identity gene *APETALA1* (*AP1*)[5]. The *VRN2* gene is a dominant repressor of flowering that is downregulated by vernalization[6] and encodes a putative zinc finger and a CCT domain (ZCCT) containing protein, which has no ortholog in Arabidopsis[7]. *VRN3* also promotes flowering and is upregulated by vernalization[8] and encodes a RAF kinase inhibitor similar to FLOWERING LOCUS T (FT) in Arabidopsis[9,10]. *VRN4* is similar to *VRN1* in sequence and function but exists in an ancient subspecies *T. aestivum* ssp. *sphaerococcum* from South Asia[11]. Each of these *VRN* genes was cloned on the basis of qualitative variation in the vernalization requirement between spring wheat and winter wheat. Winter wheat cultivars are also classified into three types according to the quantitative variation in the duration of low temperature required to reach a vernalization saturation point: weak winter types, semi-winter types, and strong winter types[12]. The positional cloning of a major quantitative trait locus (QTL) that regulates the duration of the low-temperature requirement between semi-winter and strong winter wheat cultivars revealed that the trait is controlled by recessive *vrn1* alleles encoding two different vrn1 proteins[13].

Wheat cultivars are also classified into two groups based on their photoperiod sensitivity: the photoperiod-insensitive type with early flowering and the photoperiod-sensitive type with late flowering. The photoperiod gene *PPD-H1* in barley (*Hordeum vulgare* L.) was cloned using positional cloning approach[14], and its sequence provides intuitive information on orthologous genes on three homoeologous chromosomes in common wheat (*PPD-A1*, *PPD-B1*, and *PPD-D1*). Molecular markers developed for these wheat *PPD* genes have accelerated their extensive use in breeding[15–18].

All of these cloned and characterized vernalization and photoperiod genes confer dramatic phenotypic variation. The difference in heading date can be as large as a few months between spring and winter wheat[4,6], a couple of months among photoperiod-sensitive and insensitive cultivars[14], and up to 45 days among semi-winter and strong winter cultivars tested under constant temperatures[13]. The substantial effects caused by these genes have facilitated their cloning but limited their use in a given wheat growth area because spring wheat is not well-adapted to winter wheat areas and vice versa. Allelic variation in recessive *vrn1*, recessive *vrn3*, and *PPD*-sensitive alleles is exploited to regulate the developmental phases of winter wheat cultivars, and these tri-locus genetic combinations have allowed coarse tuning of the phenotypic variation among contemporary winter wheat cultivars in the Great Plains[13,17]. However, some winter wheat cultivars possess minor differences in heading date that cannot be explained by allelic variation in these known vernalization and photoperiod alleles/genes or that can be explained only by cloning additional genes.

Recent studies have shown that mean global surface air temperature has risen and will continue to increase in the future[19,20]. Winter wheat cultivars will become increasingly vulnerable to the resulting higher temperatures and climate warming, because a decrease in the duration of low temperatures may result in no or incomplete vernalization[21–28]. Delays in flowering time caused by changes in the climate might be minor, but minor differences can be critical for sustainable food production. To date, little is known about genes that mediate those minor differences in flowering time among winter wheat cultivars.

In this work, we use a map-based cloning approach to clone a gene encoding an *O*-GlcNAc transferase that can be used to finely regulate flowering time in winter wheat cultivars.

## Results

**A QTL for heading date was identified and cloned from a winter wheat population.** The two winter wheat cultivars Duster and Billings were identified to possess the same allele for each of three known developmental genes: *vrn-A1b*, *PPD-D1b*, and *vrn-D3b* that were genotyped according to their allelic variation[13,17] (Supplementary Fig. 1) (Supplementary Method 1). However, Billings showed early development and Duster showed delayed development under greenhouse conditions without vernalization (Fig. 1a) and with vernalization (Supplementary Fig. 2a). Therefore, we sought genes responsible for the difference between the two winter wheat cultivars by phenotyping the Duster × Billings population of 260 doubled-haploid (DH) lines (Fig. 1b). This DH population was phenotyped under various environments and was genotyped using genotype-by-sequencing (GBS) markers[29]. We integrated the phenotypic data with those from the GBS markers and identified a QTL for heading date that mapped to the short arm of chromosome 6A. This QTL for heading date was thus referred to as *QHd.osu-6A* (Fig. 1c). The *QHd.osu-6A* locus explained 14.3% (LOD = 8.7), 11.7% (LOD = 7), and 10.2% (LOD = 6.1) of the total phenotypic variation in the population tested in the greenhouse (Fig. 1c). This locus also explained the subtler difference in heading date between the two alleles when the same population was tested in the field with natural vernalization (Supplementary Fig. 3). The result was confirmed in five critical DH lines without (Supplementary Fig. 2b) and with vernalization (Supplementary Fig. 2c). The co-localization of *QHd.osu-6A* for heading date from various environments revealed its consistent genetic effect in winter wheat.

Initially, we used five recombinant lines in the DH population to map the candidate genes for *QHd.osu-6A* within an ~2.1-Mb region between two flanking markers, GBS7889 and GBS10048 (Supplementary Fig. 3), according to the IWGSC RefSeq v1.0 genomic sequence of Chinese Spring (CS). Then, we developed 11 internal markers, including four regular PCR markers (Supplementary Fig. 4), three sequencing markers (Supplementary Fig. 5), and four Kompetitive allele-specific PCR (KASP) markers (Supplementary Fig. 6) to fine-map *QHd.osu-6A*. We also identified six new crossovers in the region between GBS7889 and GBS10048 (Fig. 1d) from 618 $F_{2:4}$ plants derived from a Duster × Billings cross (Supplementary Method 2). We phenotyped eleven critical lines, including five DH lines and six recombinant lines, in the presence and absence of vernalization for fine mapping (Fig. 1d). The gene responsible for *QHd.osu-6A* was narrowed down to within a 382,146-bp genomic region flanked by two markers, 6A-KASP13 and 6A400M1. According to the IWGSC RefSeq v1.0 sequences, this region contained three candidate genes: *TraesCS6A01G091100*, *TraesCS6A01G091200*,

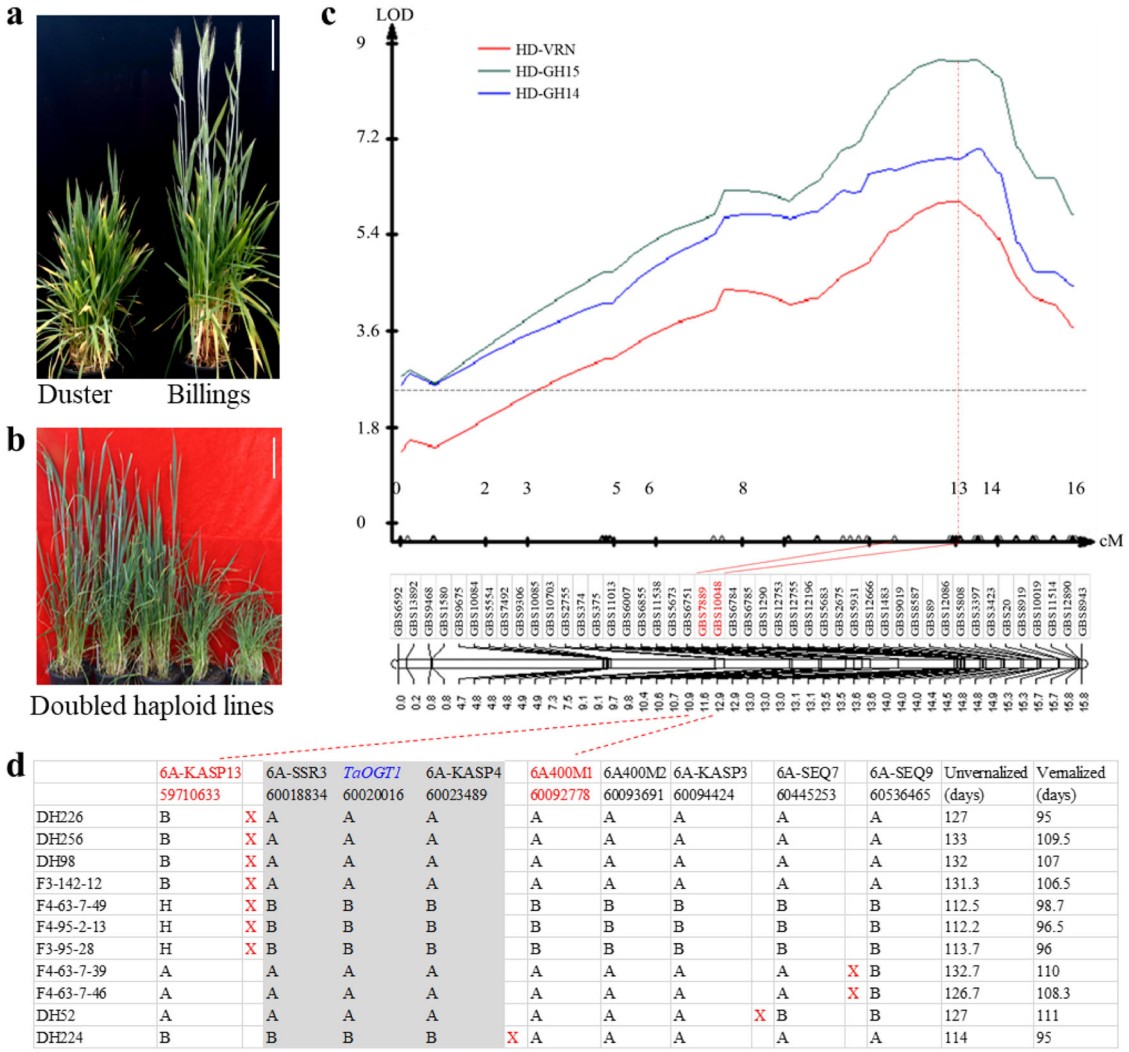

**Fig. 1 Mapping and cloning of *QHd.osu-6A*. a** The two parental lines, Duster (left) and Billings (right), grown in a greenhouse with controlled temperatures and long days and without vernalization. **b** Representative DH plants segregating for heading date in the Duster × Billings DH population in the greenhouse and without vernalization. **a, b** Scale bar = 15 cm. **c** Mapping of *QHd.osu-6A* for heading date. Phenotypic data were obtained from the DH population grown in the greenhouse and vernalized (HD-VRN) or not vernalized in 2014 (HD-GH14) and 2015 (HD-GH15). The GBS marker data were deposited in the NCBI SRA (accession number SRP051982) in a previous study[13]. Two GBS markers, GBS7889 and GBS10048, linked with the QTL peak, are indicated in red. The vertical dashed line indicates that the two GBS markers are under the QTL peaks. The *x*-axis represents genetic distance (cM) within the genomic region covering *QHd.osu-6A*, and the *y*-axis represents the LOD value of *QHd.osu-6A*. The horizontal dashed line in the QTL curve of *QHd.osu-6A* represents a threshold log of the odds (LOD) value of 2.5. **d** Fine mapping of *QHd.osu-6A* using nine internal markers to genotype five DH lines and six F$_{3:4}$ lines that contained crossovers at the targeted region. X in red indicates a crossover. Two markers flanking the candidate gene are indicated in red; three markers linked to the candidate gene are highlighted in gray. The final candidate is indicated in blue. The numbers under the markers are the physical locations (bp) of these markers according to the IWGSC RefSeq v1.0 genomic sequence. Heading date was phenotyped for days from planting to heading with six replicates for each of DH lines and 15–20 plants for in each of F$_{3:4}$ lines. Source data underlying (**c**) are provided as a Source Data file.

and *TraesCS6A01G091300* (Fig. 2a). Finally, we cloned the gene for *QHd.osu-6A* using the classic map-based cloning approach (Supplementary Method 2).

Next, we sequenced each of the two alleles, derived from Duster and Billings, respectively, for each of the three candidate genes. The sequences of *TraesCS6A01G091100* (Supplementary Fig. 7) and *TraesCS6A01G091200* (Supplementary Fig. 8), which encode pentatricopeptide repeat-containing proteins, were the same in Duster and Billings. Furthermore, the transcript levels of both genes did not differ significantly between the Duster and Billings alleles (Supplementary Fig. 9). Therefore, the two genes were excluded as candidates for *QHd.osu-6A*, and the third gene, *TraesCS6A01G091300*, was left as the sole candidate for *QHd.osu-6A*.

*TraesCS6A01G091300* consists of eight exons and seven introns and shows numerous nucleotide differences within the sequenced region between Duster and Billings (which had the same sequence as CS). These included differences within the genic region, ~1.2 kb before the start codon and 121 bp after the stop codon (Fig. 2b, Supplementary Fig. 10). One single-nucleotide polymorphism (SNP) in the coding region resulted in the substitution of a valine residue at position 229 of the protein encoded by the Duster (late) allele for the isoleucine encoded by the Billings (early) allele (Supplementary Fig. 11).

*TraesCS6A01G091300* is annotated in the IWGSC RefSeq v1.0 database to encode a transmembrane protein with 343 amino acids. When we used the predicted *TraesCS6A01G091300* protein sequence to query nonredundant protein databases in GenBank,

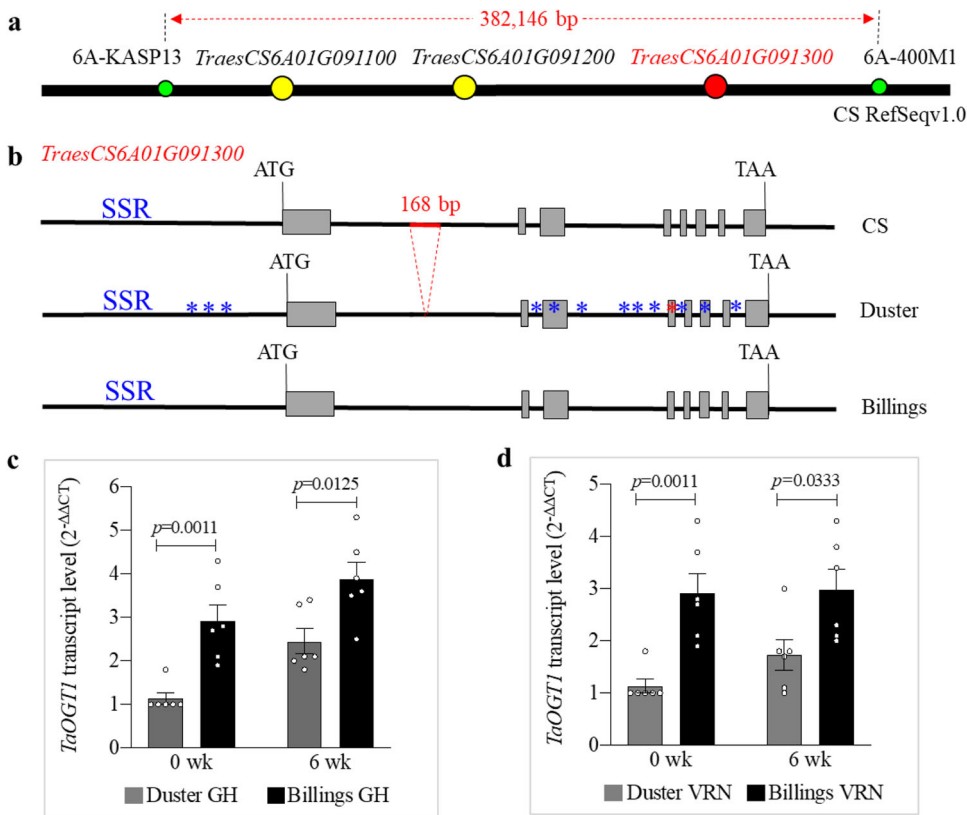

**Fig. 2 Physical location and identification of *QHd.osu-6A* candidate gene. a** The physical locations of *QHd.osu-6A* candidate genes. The *QHd.osu-6A* interval is delimited within 382,146 bp, flanked by the two markers 6A-KASP13 and 6A-400M1 that are indicated in green dots. The candidate gene, *TraesCS6A01G091300*, is indicated in a red dot, and the other two candidate genes in this region are indicated in yellow dots. **b** Allelic variation in *TraesCS6A01G091300* among Chinese Spring (CS), Duster, and Billings. The sequence of *TraesCS6A01G091300* in Billings is the same as in CS, but the sequence of Duster has SNPs across the gene, which are indicated with blue asterisks. A red asterisk indicates a SNP that changes amino acid. SSR denotes simple sequence repeats in the promoter region of *TraesCS6A01G091300*. The region of 168-bp insertion in intron 1 of the Billings allele is indicated by a red line. **c, d** Comparison of transcript levels of *TraesCS6A01G091300* alleles in Duster and Billings in non-vernalized (GH) and vernalized plants (VRN) at two different stages. The RNA samples were collected from fully developed leaves at the fourth-leaf stage (0 wk) and after vernalization for 6 weeks (6 wk). Transcript levels of *TraesCS6A01G091300* were analyzed using qRT-PCR and calculated using the $2^{-\Delta\Delta CT}$ method, where CT is the threshold cycle. The mean transcript level for each allele was analyzed using three biological replicates and two technical replicates, and a two-tailed unpaired Students' *t* test showed no significant difference in expression between the two alleles ($n = 6$). The bars indicate standard error. Source data underlying (**c**, **d**) are provided as a Source Data file.

all of its hits in plants represented hypothetical or uncharacterized proteins except for an ortholog in Arabidopsis (NP_201101), which is annotated as a putative uridine diphosphate (UDP)-*N*-acetylglucosamine-*N*-acetylmuramyl-pyrophosphoryl-undecaprenol *N*-acetylglucosamine protein (an *O*-GlcNAc transferase, abbreviated OGT) (Supplementary Fig. 11). The Arabidopsis NP_201101 is functionally uncharacterized and shows no similarity or domain conservation with the known plant OGT protein, SECRET AGENT (*At*SEC)[30]. The Arabidopsis SPINDLY (*At*SPY) was reported to be an OGT protein[31] but it was recently believed to be an *O*-fucosyltransferase[32,33]. *TraesCS6A01G091300* is referred to as *TaOGT1* in common wheat, with *TaOGT1b* representing the early allele in Billings and *TaOGT1a* representing the late allele in Duster.

**A key regulatory DNA element differentiated transcript levels of the two *TaOGT1* alleles.** We formulated two mutually exclusive hypotheses to address how the two alleles of *TaOGT1* generate differential phenotypes. One hypothesis was that the heading date was regulated by *TaOGT1* at the transcriptional level as a result of SNPs in the promoter and the non-coding regions. The alternative hypothesis was that the heading date was regulated by *Ta*OGT1 at the protein level, due to the point

mutation in this protein. Therefore, we set out to test the two hypotheses and validated that the heading date was regulated by *TaOGT1* at the transcriptional level.

We observed that the transcript level of *TaOGT1b* in Billings was significantly higher than that of *TaOGT1a* in Duster when each was grown at a variety of broad ambient temperatures either without (Fig. 2c) or with vernalization (Fig. 2d). The Billings *TaOGT1b* allele contained a 168-bp insertion in intron 1 compared to the Duster *TaOGT1a* allele (Fig. 2b, Supplementary Fig. 12). We developed two constructs to test whether the 168-bp fragment is involved in regulation of reporter gene expression in a transient GUS expression system. The construct *TaOGT1b*-Prom included 300 bp of the promoter sequence, exon 1, and 200 bp of intron 1 containing the 168-bp fragment from the Billings allele, and the *TaOGT1a*-Prom sequence included 300 bp of promoter sequence, exon 1, and 32 bp of intron 1 without the 168-bp fragment from the Duster allele (Fig. 3a). The two constructs contained the identified 300 bp promoter sequence excluding the SNPs in the promoter between the two *TaOGT1* alleles and had the only difference for the presence and absence of the 168-bp fragment. The *LUC* gene was driven by the ubiquitin promoter, which was used as an internal transformation control to provide an estimate of the efficiency of transient expression. We co-

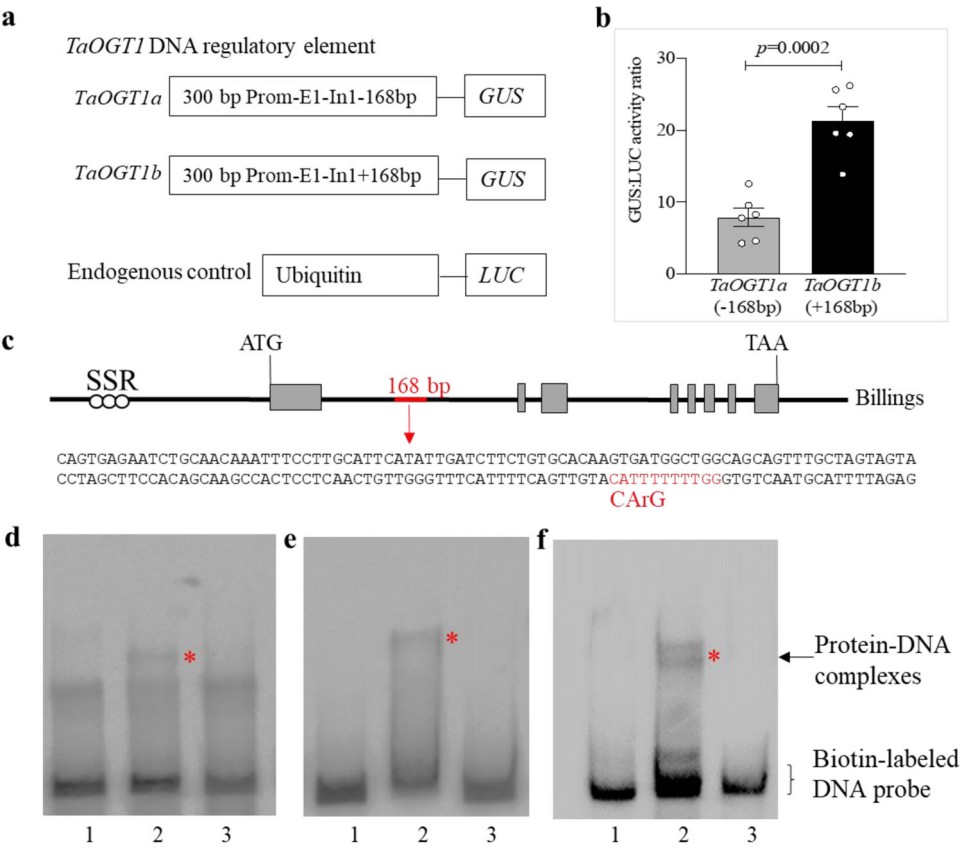

**Fig. 3 Characterization of *TaOGT1*. a** A diagram depicting the constructs used to test regulatory elements in *TaOGT1*. The promoter (Prom), exon 1 (E1), and part of intron 1 (In1) from Duster *TaOGT1a* without the 168-bp fragment, and from Billings *TaOGT1b* with the 168-bp fragment, were used as a promoter of the *GUS* reporter gene. The endogenous control construct was created using the *Ubiquitin* promoter to drive the *LUC* gene. **b** Comparison of GUS/LUC activity ratio. The GUS and LUC activities were assayed in a transient expression in wheat protoplast system. The data are presented as mean ratio ± SEM, which was calculated from six independent protoplast transformation reactions ($n = 6$) for each construct. A two-tailed unpaired $t$ test showed significant difference between the two constructs from the two *TaOGT1* alleles. **c** The sequence of the 168-bp insertion in intron 1 of the Billings *TaOGT1b* allele. The CArG box is indicated in red. **d–f** EMSA interactions of the vrn1 and *Ta*VRT2 proteins with the CArG-box DNA probe labeled with biotin. EMSA was performed for the vrn1b protein from 2174 (which has the same vrn1b as Billings and Duster) (**d**), vrn1a from Jagger (**e**), and *Ta*VRT2 (**f**). Lane 1 indicates the biotin-labeled *TaOGT1b* DNA probe alone. Lane 2 represents vrn1 or *Ta*VRT2 protein and DNA in the EMSA reactions. Lane 3 represents vrn1 or *Ta*VRT2 protein and 100× protein competitors that were added to the EMSA reactions. These EMSA reactions were repeated three times. Source data underlying (**b**, **d–f**) are provided as a Source Data file.

transformed *TaOGT1b*-Prom and *TaOGT1a*-Prom, respectively, with the LUC construct into wheat protoplasts to assess their promoter activity. Protoplasts transformed with *TaOGT1b*-Prom showed much higher GUS expression than protoplasts transformed with *TaOGT1a*-Prom, which lacks the 168-bp fragment (Fig. 3b). This result provided experimental evidence that the 168-bp insertion/deletion (indel) was important for the transcriptional regulation of *TaOGT1*.

We analyzed the sequence of the 168-bp DNA fragment of the *TaOGT1b* allele and identified a potential CArG-box (CATTTTTTGG) (Fig. 3c), which is a conserved DNA target site of MADS-domain proteins including VRN1[4,34–36] and *Ta*VRT2[37–39]. In an in vitro electrophoretic mobility shift assay (EMSA), both vrn-A1 (Fig. 3d, e) and *Ta*VRT2 (Fig. 3f) proteins physically bound the 168-bp DNA fragment containing this CArG-box. We concluded that the Billings *TaOGT1b* allele has a direct target site for the MADS-box proteins in wheat, but the Duster *TaOGT1a* allele does not have this target site.

**Constitutive overexpression of the Billing *TaOGT1b* allele in Duster accelerated heading of transgenic plants.** We performed a transgenic complementation analysis to confirm that *TaOGT1b* from Billings accelerated flowering in transgenic Duster plants.

We transformed the *TaOGT1b* allele into immature Duster embryos through particle bombardment of an expression construct in which the maize *Ubiquitin* promoter was fused with *TaOGT1b* in the expression vector pMDC32. This generated seven positive T0 plants, two of which (*TaOGT1b*-OE9 and *TaOGT1b*-OE12) were self-pollinated to produce T1 seeds for further studies. In each of the two T1 populations, *TaOGT1b* genetically segregated as a single-copy transgene. On average, the heading dates of transgenic plants were 21 days earlier in the unvernalized populations (Fig. 4a, b) and 6 days earlier in the vernalized population (Fig. 4c, d) than that of the non-transgenic sibling populations.

**TaOGT1 had the O-GlcNAcylation enzyme activity.** We characterized the OGT activity in the *TaOGT1b*-overexpressing transgenic plants. First, we tested whether *Ta*OGT1 had similar activity to O-GlcNAc transferase, using a UDP detection kit. In this OGT detection reaction, a GlcNAc group from the UDP-GlcNAc donor is transferred to the side chain of serine and threonine on the OGT substrates, and the released UDP can be detected to indicate OGT activity (Fig. 4e). We attempted to express the full-length *Ta*OGT1 protein in *Escherichia coli* for this experiment, but were unsuccessful, probably because it is a

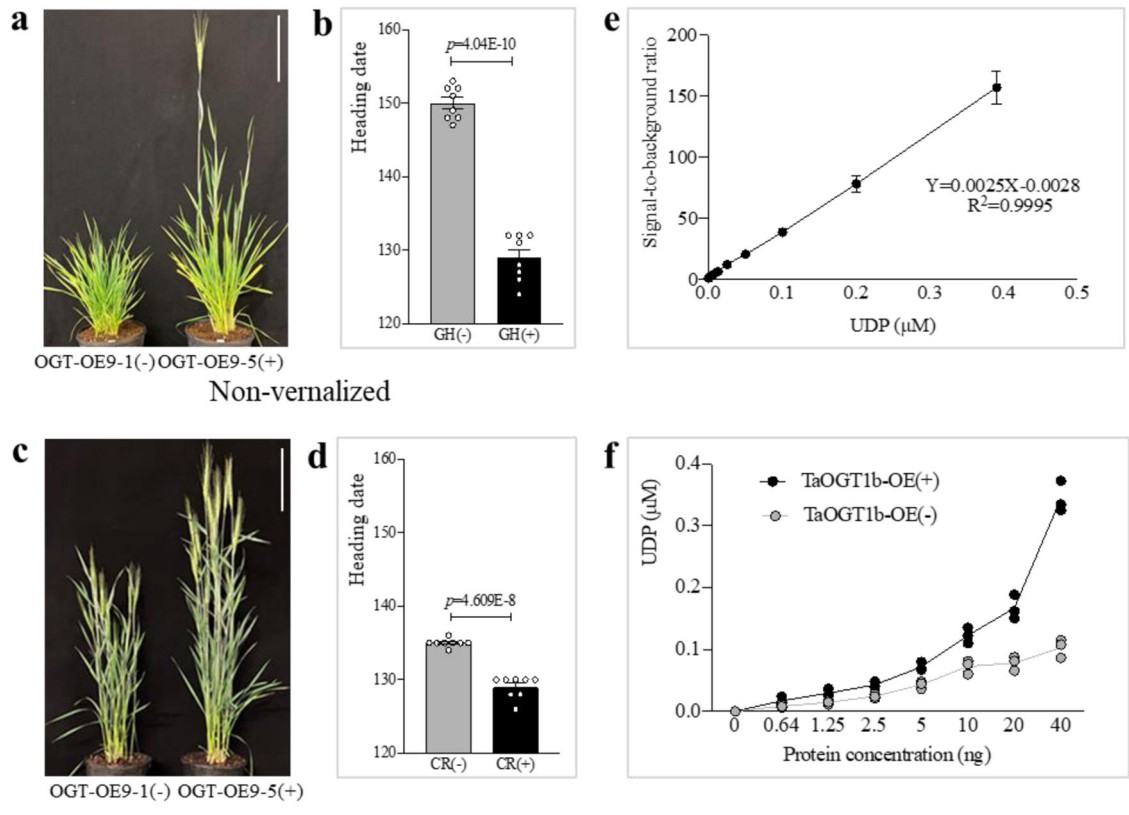

**Fig. 4 Constitutive overexpression of *TaOGT1b* and its enzyme activity. a** *TaOGT1b* from Billings was overexpressed in Duster. A transgenic (right) and a non-transgenic plant (left) grown in the greenhouse without vernalization are shown. **b** Comparison of the mean heading date for transgenic and non-transgenic plants in the two $T_1$ populations without vernalization. **c** A transgenic (right) and a non-transgenic plant (left) grown in the greenhouse with vernalization. **a**, **c** Scale bar = 15 cm. **d** Comparison of the mean heading date for transgenic and non-transgenic plants in the two $T_1$ populations with vernalization. **b**, **d** Statistical comparison of the mean heading date was performed using a two-tailed unpaired Student's *t* test to determine the significance level between transgenic and non-transgenic plants ($n = 8$). The error bars indicate standard error. **e** A UDP standard curve for the amount of UDP (*x*-axis) and the signal-to-background ratio (*y*-axis) based on the luminescence reading in the samples ($n = 3$). **f** Comparison of *TaOGT1b* enzyme activity in the total protein extract from the leaves of *TaOGT1b-OE9* transgenic and non-transgenic plants. The *x* and *y* axes represent the amounts of protein (ng) and UDP (calculated from the UDP standard curve) in the samples ($n = 3$), respectively. Source data underlying (**b**, **d–f**) are provided as a Source Data file.

eukaryotic-membrane-like protein that is not expressed in *E. coli*[40]. Therefore, we extracted total protein from transgenic *TaOGT1b-OE9* plants, which included overexpressed *TaOGT1b* and native *TaOGT1a*, and total protein from non-transgenic plants, which contained only native *TaOGT1a*. Protein extract from transgenic *TaOGT1b*-OE9 plants showed higher *O*-GlcNAc transfer activity than that extracted from non-transgenic plants (Fig. 4f). Using 20 ng of each total protein sample, we detected UDP concentrations of 0.167 μM in *TaOGT1b*-OE9 wheat plants but only 0.078 μM UDP in non-transgenic plants. Together, these results demonstrate that *TaOGT1* has *O*-GlcNAcylation enzyme activity.

***TaOGT1 O*-GlcNAcylated *Ta*GRP2**. To identify the substrates of the *TaOGT1* protein, *TaOGT1b* was used as a bait to screen a yeast-two hybrid (Y2H) library established using whole seedlings of winter wheat cultivar "2174"[41]. From the screening of ~$2 \times 10^7$ cells, we identified two interacting proteins, both of which were annotated as nonspecific serine/threonine protein kinases: *Traes*CS1B02G364800, hereafter referred to *Ta*K1 (Supplementary Fig. 13), and *Traes*CS4D02G196100, hereafter referred to *Ta*K4 (Supplementary Fig. 14). We confirmed the direct interaction between *TaOGT1b* and *Ta*K1 or *Ta*K4 by cotransformation of yeast cells (Supplementary Fig. 15) and using a transient expression system in tobacco leaves (Supplementary

Fig. 16), but neither was a direct substrate of *O*-GlcNAcylation by *TaOGT1* in vitro (Supplementary Fig. 17).

The *Ta*GRP2 protein (*Traes*CS4B02G020300) (Supplementary Fig. 18) is reported to be a glycine-rich RNA-binding protein (GR-RBP or GRP) that is *O*-GlcNAcylated during vernalization in wheat[42]. The direct interaction between *TaOGT1b* and *Ta*GRP2 was observed in co-transformation of yeast cells (Supplementary Fig. 15) and in a transient expression system with bimolecular fluorescence complementation (BiFC) in tobacco leaves (Supplementary Fig. 16). We found that proteins extracted from *TaOGT1b* transgenic plants produced a higher *O*-GlcNAcylation signal of *Ta*GRP2 than those from non-transgenic plants (Fig. 5a, Supplementary Fig. 19a), indicating that *TaOGT1* could *O*-GlcNAcylate *Ta*GRP2. The amount of GlcNAcylated *Ta*GRP2 proteins by *TaOGT1* in the GlcNAcylation reaction with *Ta*K1 was significantly lower than that in the control reaction without *Ta*K1 (Fig. 5b, Supplementary Fig. 19b), indicating that the GlcNAcylation of *TaOGT1* on *Ta*GRP2 was repressed by *Ta*K1. However, *Ta*K4 showed no effect on the GlcNAcylation of *TaOGT1* on *Ta*GRP2 (Fig. 5b). Surprisingly, the *TaOGT1* protein was able to GlcNAcylate some *E. coli* proteins that were not purified from the *Ta*GRP2, *Ta*K1, or *Ta*K4 protein samples (Fig. 5b, Supplementary Fig. 17), indicating that *TaOGT1* has the capacity of GlcNAcylation on diverse proteins in different organisms.

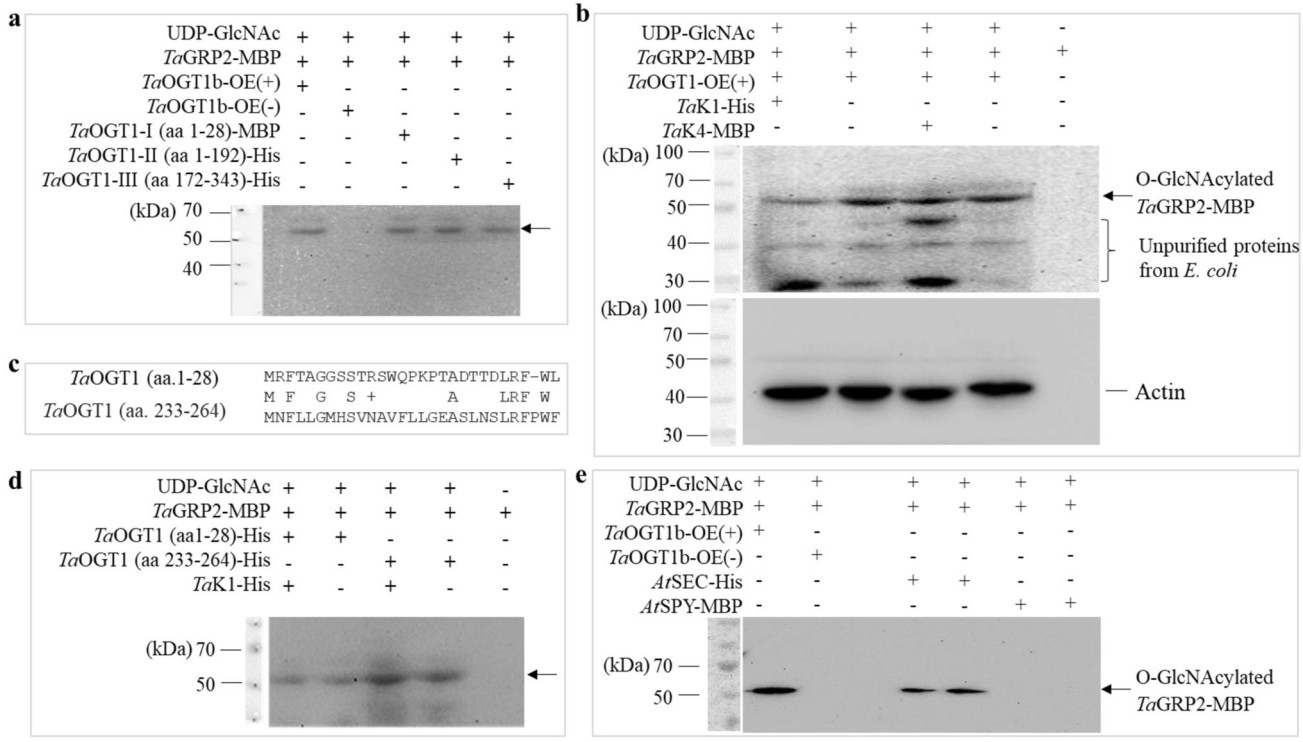

**Fig. 5 GlcNAcylation of _Ta_OGT1 in vitro. a** Comparison of OGT enzyme activity among different _Ta_OGT1 proteins. _Ta_OGT1 proteins from a transgenic plant (+), _Ta_OGT1 proteins from a non-transgenic plant (−), and three different _Ta_OGT1 protein fragments expressed in _E. coli_: aa 1–28, aa 1–192, and aa 172–343. Presence and absence of the particular protein in the reaction is indicated by "+" and "−", respectively. The GlcNAcylated _Ta_GRP2 protein was detected using the anti-_O_-GlcNAc antibody CTD110.6 and is highlighted by an arrow. **b** Effects of _Ta_K1 and _Ta_K4 on GlcNAcylation by _Ta_OGT1. _Ta_OGT1 was extracted from total protein from leaves of transgenic wheat plants, and _Ta_K1 or _Ta_K4 were added to the GlcNAcylation reaction. The upper image indicates GlcNAcylated _Ta_GRP2 protein detected with the CTD110.6 antibody against _O_-GlcNAc, and the lower image indicates actin protein as an endogenous control, detected by an anti-actin antibody. **c** Comparison of the sequences of two GlcNAcylation sites. **d** Comparison of OGT enzyme activity of two protein fragments of aa 1–28 and aa 233–264. **e** Comparison of _At_SEC and _At_SPY proteins expressed in _E. coli_ with _Ta_OGT1 proteins from a transgenic plant (+) and a non-transgenic plant (−). _Ta_GRP2 protein was used as a substrate in each GlcNAcylation reactions. The images of the same gels stained with Coomassie Brilliant Blue (CBB) are provided in Supplementary Fig. 19. These GlcNAcylation reactions were repeated three times. Source data underlying (**a**, **b**, **d**, **e**) are provided as a Source Data file.

**_Ta_OGT1 protein had two active sites.** We found that the _Ta_OGT1 protein contains two active sites for _O_-GlcNAcylation. We split _Ta_OGT1 into three protein fragments, _Ta_OGT1-P1 (aa 1–28), _Ta_OGT1-P2 (aa 1–192), and _Ta_OGT1-P3 (aa 172 to the final residue, aa 343), to facilitate their expression in _E. coli_. Notably, all three fragments could GlcNAcylate _Ta_GRP2 (Fig. 5a). Sequence analysis suggested that a conserved domain similar to aa 1–28 in _Ta_OGT1-P1 was present at aa 234–264 in _Ta_OGT1-P3 (Fig. 5c, Supplementary Fig. 20). Therefore, we expressed the _Ta_OGT1-P4 fragment (aa 234–264) and found that both _Ta_OGT1-P1 and _Ta_OGT1-P4 catalyzed the GlcNAcylation of _Ta_GRP2 (Fig. 5d, Supplementary Fig. 19c), demonstrating that _Ta_OGT1 contains two active sites. We observed no significant difference in the level of _Ta_GRP2 _O_-GlcNAcylation by _Ta_OGT1a and _Ta_OGT1b, despite one amino acid difference between their protein sequences (Supplementary Fig. 21). Therefore, the difference in heading date between Duster and Billings was not caused by _Ta_OGT1 at the protein level.

In Arabidopsis, _At_SEC is reported to have the _O_-GlcNAcylation enzyme activity[30], and _At_SPY is reported to have the _O_-fucosyl-transferase activity[33,34] but not the _O_-GlcNAcylation enzyme activity[31]. We found that _Ta_GRP2 was GlcNAcylated by _At_SEC but not by _At_SPY (Fig. 5e, Supplementary Fig. 19d). The results indicated that although _Ta_OGT1 and _At_SEC have no similarity or domain conservation in protein sequences, they showed a similar GlcNAcylation activity on the same substrate as _Ta_GRP2.

**_Ta_OGT1 activity was associated with sugar content and the transcript levels of flowering time genes.** To establish a signaling network, we tested which genes and metabolites were regulated in _Ta_OGT1b-overexpressing transgenic plants, using non-transgenic plants as controls. We tested these plants in the greenhouse without vernalization and with vernalization for 1, 3, or 6 weeks. Both the transcript levels of _Ta_OGT1 (overexpressing _Ta_OGT1b from Billings plus native _Ta_OGT1a in Duster) (Fig. 6a) and the enzymatic activity of _Ta_OGT1 (Fig. 6b) increased in the _Ta_OGT1b-overexpressing transgenic plants. Due to the elevated transfer of GlcNAc to nucleotides as a result of the increased _Ta_OGT1 content, the content of glucose (Fig. 6c) and sucrose (Fig. 6d) decreased, but the content of fructose did not significantly change (Fig. 6e). Surprisingly, the transcript levels of _vrn1_ (Fig. 6f), _vrn3_ (Fig. 6g), _PPD1_ (Fig. 6h), and _Ta_GRP2 (Fig. 6i) all increased in the _Ta_OGT1b-overexpressing transgenic plants, compared with the non-transgenic plants. The transcript levels of either _VRN2_ (Fig. 6j) or _TaVRT2_ (Fig. 6k) did not change significantly in the _Ta_OGT1b-overexpressing transgenic plants, compared with the non-transgenic plants. The effects of low temperature on the transcript levels of these genes tested in the same samples indicated that _vrn1_ (Fig. 6f) and _vrn3_ (Fig. 6g) were significantly upregulated, _VRN2_ (Fig. 6j) and _TaVRT2_ (Fig. 6k) were significantly downregulated, but _TaGRP2_ (Fig. 6i) was not significantly regulated by low temperature. The downregulation of _TaOGT1_ by low temperature was detectable in the 3 weeks-

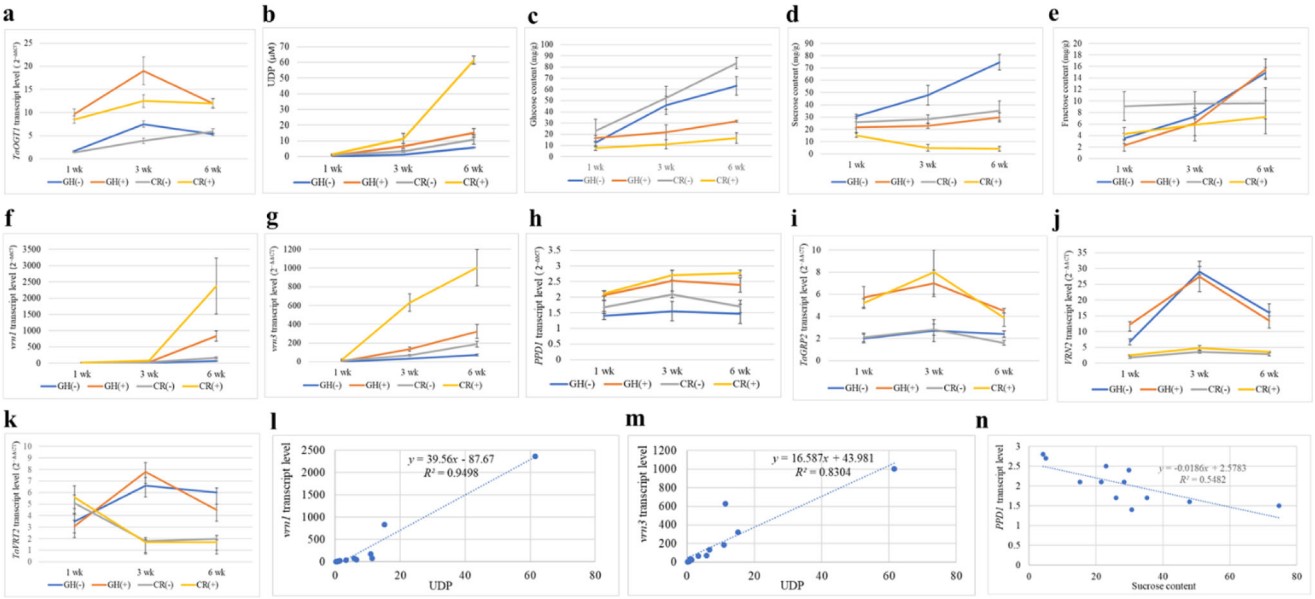

**Fig. 6 Effects of *TaOGT1b* overexpression on sugar content and flowering-time gene expression. a–k** Comparison of gene expression and sugar content in *TaOGT1b* overexpressed transgenic plants (+) and non-transgenic plants (−). The plants were grown in the greenhouse (GH) without vernalization or were treated in a cold room (CR) with vernalization for 1 week (1 wk), 3 weeks (3 wk), or 6 weeks (6 wk). The same samples of fully developed leaves from three different plants with two technical replicates ($n = 6$) were used to determine *TaOGT1* transcript levels (**a**); *O*-GlcNAc transferase activity (**b**); the contents (mg g$^{-1}$) of glucose (**c**), sucrose (**d**), and fructose (**e**); and transcript levels of *vrn1* (**f**), *vrn3* (**g**), *PPD1* (**h**), *TaGRP2* (**i**), *VRN2* (**j**), and *TaVRT2* (**k**). Transcript levels were calculated using the $2^{-\Delta\Delta CT}$ method, where CT is the threshold cycle. The significant differences in mean expression for each gene were compared between the two alleles using a two-tailed unpaired Student's $t$ test (Supplementary Table 1); the bars indicate the standard error. **l–n** Correlation analyses of data for all transgenic and non-transgenic plants with and without vernalization ($n = 12$): l. UDP *vs. vrn1*; m. UDP *vs. vrn3*; **n** sucrose *vs. PPD1*. The significant correlations between two variables were shown in Supplementary Tables 2, 3. Source data underlying (**a–k**) are provided as a Source Data file.

vernalized plants but not in the 6 weeks-vernalized plants (Fig. 6a). The significance levels of each gene and sugar regulated by *TaOGT1* and low temperature are summarized in Supplementary Table 1. We analyzed the association of *TaOGT1* enzymatic activity and *TaOGT1* transcript level with content of each sugar and transcript level of each flowering-time gene in all transgenic and non-transgenic plants with and without vernalization (Supplementary Tables 2, 3). The most striking correlations included a positive correlation between *TaOGT1* activity and transcript levels of *vrn1* ($R^2 = 0.9498$, $p < 0.00001$, Fig. 6l) and *vrn3* ($R^2 = 0.8304$, $p < 0.0001$, Fig. 6m). In addition, there was a negative correlation between sucrose content and *PPD1* transcript levels ($R^2 = 0.5482$, $p = 0.0058$, Fig. 6n). The results indicated that *TaOGT1* enzymatic activity was associated with the transcript levels of the flowering-time genes and sugar content in the cytoplasmic matrix of wheat leaves.

## Discussion

In this study, we cloned *TaOGT1* and revealed the signaling networks that regulate heading date in winter wheat (Fig. 7). *TaOGT1b* promoted the heading of winter wheat cultivars under broad ambient temperatures because its intron contains a binding site for MADS-box transcriptional factors, including VRN1 and *Ta*VRT2, which were tested in this study. *VRN1* is a central regulator of heading date in different wheat species, subspecies, and cultivars[4,13,36,43–45]. *Ta*VRT2 was reported to be a repressor of wheat development, whose transcript levels were decreased by vernalization[37,38], but *Ta*VRT2 was recently reported to be a promoter of wheat development, whose transcript levels were increased[39]. We tested the transcript levels of *TaVRT2* and found that it was dramatically decreased by vernalization (Fig. 6k).

*Ta*OGT1 *O*-GlcNAcylates *Ta*GRP2, which is a negative regulator of flowering via its direct binding to *VRN1* pre-mRNA[42,46]. The *O*-GlcNAcylated modification on *Ta*GRP2 is induced by vernalization[46]. Our results indicated that *TaGRP2* was not significantly regulated by low temperature, but it was dramatically increased in the *TaOGT1* overexpressed transgenic wheat plants, compared with non-transgenic plants (Fig. 6i). Due to the *O*-GlcNAcylation of *Ta*OGT1 targets, the content of soluble glucose decreased, which promoted the expression of *vrn1*, *vrn3*, and *PPD1*. Therefore, *Ta*OGT1 is a critical regulator of flowering in winter wheat cultivars.

Previous studies showed that *vrn1* genes are induced by low temperature in winter wheat[4,8,43–45]. The upregulation of *vrn1* is due to indels in the promoter region that result in the loss of a recognition site by repressors[4,6,36,47,48] or the gain of a binding site by the microRNA *Ta*miR1123[49]. The repressor binding sites are also located within *vrn1* intron 1[50] and the 5′ untranslated region[48]. The upregulation of *vrn3* by low temperature is due to a large insertion in the promoter region[6]. Overexpression of *Ta*OGT1 potentially modifies the proteins involved in the interactions with the regulatory sites of these key flowering genes in wheat.

In Arabidopsis, *At*SEC was identified based on their sequence similarity to OGTs in animals[30]. Although *At*SEC is different in sequence from *Ta*OGT1, *At*SEC was able to *O*-GlcNAcylate *Ta*GRP2. There are up to 34 unannotated or uncharacterized proteins in Arabidopsis that share similarity with wheat *Ta*OGT1. The cloning and characterization of *TaOGT1* suggest that orthologous and homologous proteins such as NP_201101 in Arabidopsis might be OGTs and may be involved in the *O*-GlcNAcylation of 262 proteins that are modified by *O*-GlcNAc in Arabidopsis[51]. There are 168 *O*-GlcNAc-modified proteins in

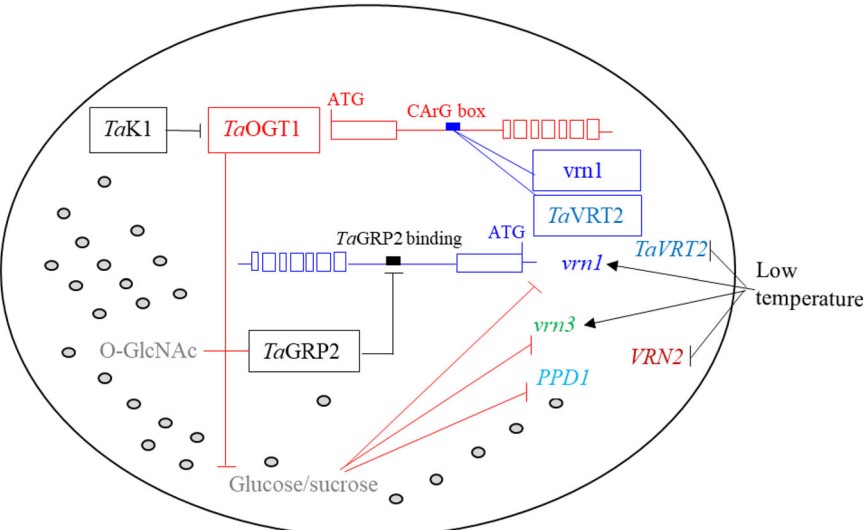

**Fig. 7 A model for the regulation of heading date by the *Ta*OGT1-sugar pathway in winter wheat.** Arrows indicate promotion and "—|" indicates repression. The genotypes of the winter wheat cultivars Billings and Duster contain recessive *vrn1* and *vrn3* alleles and a photoperiod-insensitive allele, *PPD1*. Italic names indicate genes, and non-italic names indicate proteins. Shaded gray circles represent the cytoplasmic matrix.

wheat[46], and as the characterized OGT in wheat, *Ta*OGT1 can be used to reveal functions of the homoeologous and homologous OGTs in this species. Sugar may be involved in the transition from the vegetative to reproductive phase in wheat, based on the observation that the levels of monosaccharides such as glucose and fructose are significantly higher in cold-treated plants, as a result of starch degradation and sucrose metabolism[52,53]. However, evidence for a direct role of sugar in flowering is limited. Our findings provide experimental evidence that the transcript levels of *vrn1*, *vrn3*, *PPD1*, and *TaGRP2* were upregulated by *TaOGT1* overexpression in transgenic wheat. It is likely that the upregulation of these different flowering-time genes is due to the resulting reduced glucose content in the common cytoplasmic matrix in wheat leaves. In humans and animals, only 2–5% of all glucose is used to generate the UDP-GlcNAc sugar nucleotide, but *O*-GlcNAc can serve as a nutrient and stress sensor[54,55]. In Arabidopsis, trehalose-6-phosphate (T6P), which is present only in trace amounts, functions as a proxy for carbohydrate status to regulate flowering, and when the gene encoding T6P synthase 1 (TPS1) was knocked down, plants flowered extremely late[56]. Based on sequence identity, orthologous wheat genes of Arabidopsis *TPS1* (AT1G78580) can be found (e.g., ACI16353 and KAF6991139), but none of them has been functionally characterized. The identification and characterization of *TaOGT1* might represent a connection between wheat and Arabidopsis in carbohydrate signaling pathways via glucose catabolism. TPS1 catalyzes the formation of T6P from glucose-6-phosphate and UDP–glucose (UDPG)[57], whereas OGT1 catalyzes the transfer of GlcNAc from UDP-GlcNAc to serine and threonine residues of proteins[58]. Both UDPG and UDP-GlcNAc are derived from glucose catabolism. In addition, the T6P pathway also regulates flowering via the florigen *FT*. The T6P pathway regulates flowering at two sites in the plant; in the leaves where *FT* expression is induced by TPS1, and at the shoot apical meristem, where flowering-time genes are regulated by TPS1[56].

*TaOGT1* confers relatively minor effects on heading date compared with genes in vernalization and photoperiod pathways, but these minor effects nonetheless are critical for cultivars grown in the same growth area. Minor effects could also result from earliness per se (*Eps*) genes, such as *Eps-3A*[m59] or *Eps-A*[m160] in the diploid einkorn wheat *Triticum monococcum*, but no *Eps* gene has been cloned in common wheat. Fine selection for a gene with

minor effects represents a challenge in conventional breeding. The presence of a constitutive 168-bp insertion in the *TaOGT1b* Billings allele is an intrinsic factor that is independent of the environment. However, it represents an instrumental molecular marker that fine-regulates flowering time and accelerates heading date in winter wheat cultivars by several days, to potentially help them adapt to future climate scenarios. The *TaOGT1* gene would be a starting point to reveal functions of numerous orthologous and homologous proteins in flowering time and sugar metabolism in plant species.

## Methods

**Mapping and cloning of *QHd.osu-6A*.** For gene mapping, we generated a doubled-haploid (DH) population of 260 lines from a cross between Duster (PI 644016) and Billings (PI 656843), two hard red winter wheat cultivars that are cultivated extensively across the southern Great Plains of the USA. Detailed information about the two cultivars and the DH population is provided in Supplementary Method 1. In previous studies, the Duster × Billings DH population was tested in field conditions at Stillwater Research Station, Oklahoma, for 2 years, and heading date was scored when a first head completely emerged from the boot in a plant. The data for heading date were analyzed in this study. The same DH population was grown in a greenhouse at day/night temperatures of 25/20 °C with long day conditions (16 h light/8 h darkness), and the population was either vernalized (Supplementary Method 2) or not vernalized. The phenotypes of plants in these two different treatments were integrated with GBS markers, which resulted in mapping *QHd.osu-6A* to the short arm of chromosome 6A, in a region that was not reported to affect flowering time; therefore, *QHd.osu-6A* was cloned using a positional cloning approach. The primers used for molecular markers are listed in Supplementary Table 4. Detailed procedures used to clone *QHd.osu-6A* are provided in Supplementary Method 2.

**Generation of transgenic *TaOGT1b* wheat.** Primers *Ta*OGT1b-attB1-F1 and *Ta*OGT1b-attB1-R1 (Supplementary Table 5) were designed to amplify the complete cDNA of *TaOGT1b* from Billings, and the cDNA was cloned into the pMDC32 vector containing the maize *Ubiquitin* promoter[61], which has previously been successfully used to express genes in wheat[6,62]. The ubiquitin-*TaOGT1b*-pMDC32 construct was transformed into Duster by particle bombardment using a published protocol[63]. Seven individual transgenic Duster plants (T0) were generated, and two of these (*TaOGT1b*-OE9 and *TaOGT1b*-OE12) were propagated to give T1 populations for further experiments.

In an independent experiment, embryos from *TaOGT1b*-OE-9 T0 were dissected and grown in rooting medium at 22–23 °C in long days (16 h light/8 h darkness) for 2 weeks before the resulting seedlings were transferred to the soil for greenhouse analyses. At the seventh-leaf stage, 10 *TaOGT1b*-OE9 T1 plants were divided into two equal groups, of which one was vernalized and the other was not. After 1, 3, or 6 weeks of vernalization, the first complete leaf of each tiller was collected for analysis of gene expression, sugar content, and *O*-GlcNAcylation enzyme activity. The samples were collected at 17:00 h before additional lights were

turned on at 18:00 h to make a long day condition, and the central part of the fresh leaves was analyzed as recommended[64,65].

**Transient promoter activity assays**. The DNA fragments consisting of the promoter, exon 1, and intron 1 DNA from *TaOGT1a* and *TaOGT1b* were respectively fused to the *uidA* gene encoding GUS as a reporter, which were cloned into the pMDC32 vector. The primers used for cloning were GUS-HindIII-F1 and GUS-KpnI-R1 (Supplementary Table 5). The only difference between the DNA fragments was that the Billings *TaOGT1b* allele contained a 168-bp fragment in intron 1. The internal control for the *TaOGT1::GUS* assay consisted of the pMDC32 construct that contained the *Ubiquitin* promoter fused to the gene for luciferase (LUC) that was cloned using primers LUC attB1-F1 and attB1-R1 (Supplementary Table 5). This *Ubiquitin::LUC* construct was co-transformed with *TaOGT1a-GUS* and *TaOGT1b-GUS* into wheat protoplasts using PEG (40% PEG, 0.2 M mannitol and 100 mM CaCl$_2$)[66], and the GUS/LUC ratios were used to generate relative promoter activities[67]. The co-transformed protoplasts were incubated for 48 h at 25 °C in a reaction with 1 mM 4-methylumberlliferyl-β-D-glucuronide in lysis buffer (Sigma), and the reaction was terminated with 0.2 M Na$_2$CO$_3$ after 30 min. The GUS activity was assayed using a Synergy H1 reader (BIO-TEK Instruments. Winooski, VT), and LUC activity was assayed with the Luciferase 1000 Assay system (Promega, E4550). Ratios of GUS to LUC [GUS/LUC = (GUS-30 min − GUS-0 min) × 10/LUC] activities were used to define promoter activity[67].

**Quantification of gene expression**. Quantitative real time PCR (qRT-PCR) was used to determine the transcript levels of *TaOGT1*, *vrn1* (primers vrn1-F1 and vrn1-R2), *VRN2* (primers VRN2-F and VRN2-R), *vrn3* (primers vrn3-F and vrn3-R), *PPD1* (primers PPD-F1 and PPD-R1), *TaVRT2* (primers *Ta*VRT2-F and *Ta*VRT2-R), *TaGRP2* (primers *Ta*GRP2-F and *Ta*GRP2-R), and *Actin* (primers Actin-F2 and Actin-R) from the same cDNA samples. The primers for the expression of these genes are listed in Supplementary Table 5. The transcripts of *TaOGT1* in transgenic Duster plants included native *TaOGT1a* and transgenic *TaOGT1b* from Billings, whereas non-transgenic plants contained native *TaOGT1a*. Total RNA was extracted from leaves, qRT-PCRs were performed for gene expression, and detailed information is provided in Supplementary Method 3. Gene transcript levels were calculated by the 2$^{-\Delta\Delta CT}$ method, where CT is the threshold cycle.

**Electrophoretic mobility shift assay**. Primers *Ta*OGT1-168Ind-F1 and *Ta*OGT1-168Ind-R1 (Supplementary Table 5) were used to amplify the gDNA fragment from Billings using Phusion High-Fidelity DNA Polymerase (New England Biolabs, Ipswich, MA). The DNA fragment was used as a probe after labeling the 3′-OH end of the double-stranded DNA with Biotin using a Pierce™ Biotin 3′ end DNA labeling kit (Thermo Fisher Scientific, Waltham, MA). MBP-vrn-A1a protein from cultivar Jagger (aa 1–180) and MBP-vrn-A1b from cultivar 2174 (aa 1–180) were used for the electrophoretic mobility shift assay (EMSA)[13]. *Ta*VRT2 was also cloned into pMAL-C2 vector using primers (primers *Ta*VRT2-EcoRI-F1 and *Ta*VRT2-BamHI-R1) (Supplementary Table 5), and expressed MBP-*Ta*VRT2 proteins in *E. coli* (Supplementary Method 4) were used to perform electrophoretic mobility shift assay (EMSA) with the Light Shift Chemiluminescent EMSA Kit (Thermo Fisher Scientific, Waltham, MA).

**TaOGT1 enzyme activity assay**. The full-length cDNA of *TaOGT1* was cloned into different expression vectors including pSUMO and pMAL-C2, but its protein could not be produced in *E. coli*, probably because it is a membrane-like protein that is not expressed well in this species[40]. Instead, total protein was extracted from leaves of transgenic *TaOGT1b-OE9* wheat plants, which overexpressed *TaOGT1b* and expressed native *TaOGT1a*, and from wild-type plants, according to Rubio et al.[68].

*TaOGT1* activity was assayed using two approaches. First, *O*-GlcNAc transferase activity was measured using the UDP-Glo™ Glycosyltransferase detection kit (Promega, Madison, WI). In an OGT detection reaction, GlcNAc from a UDP-GlcNAc donor is transferred to serine and threonine side chains on the substrates of the OGT, which releases UDP that can be detected to infer the activity of the OGT. The reagent converts the UDP to ATP and generates light that is detected using a luminometer Synergy H1 (BioTek Instruments, Winooski, VT). The luminescence was converted to UDP concentration using a UDP standard curve. Second, chemiluminescent signals were imaged using a Fluor Chem E imaging system with a CCD camera (Protein Simple, Santa Clara, CA). After a reaction in the GlcNAcylation reaction solution (12.5 mM MgCl$_2$, 50 mM Tris–HCl, pH 7.5, and 1 mM DTT), proteins were separated by SDS–PAGE gel electrophoresis and transferred to a PVDF membrane (Bio-Rad, Hercules, CA). Immunoblotting using an antibody against *O*-GlcNAc (CTD110.6 mouse mAb HRP conjugate, Cell Signaling Technology, Danvers, MA) was used to detect *O*-GlcNAc modification of proteins[46], and an antibody to actin (Abclonal, Woburn, MA) was used as a control.

Four truncated proteins of *Ta*OGT1 were expressed in *E. coli* BL21 to determine different *O*-GlcNAcylation active sites: (1) aa 1–28 (using primers *Ta*OGT1-EcoRI-F1 and *Ta*OGT1-BamHI-R2); (2) aa 1–192 (using primers *Ta*OGT1-NdeI-F1

and *Ta*OGT1-BamHI-R3); (3) aa 172–343 (using primers *Ta*OGT1-NdeI-F2 and *Ta*OGT1-BamHI-R1); and (4) aa 233–264 (using primers *Ta*OGT1-NdeI-F5 and *Ta*OGT1-BamHI-R5). These *Ta*OGT1 peptides were tested for their ability to GlcNAcylate *Ta*GRP2, *Ta*K1, and *Ta*K4 proteins. The full-length cDNA of *TaGRP2* (using primers *Ta*GRP2-EcoRI-F1 and *Ta*GRP2-BamHI-R1), *TaK1* (using primers *Ta*K1-NdeI-F1 and *Ta*K1-BamHI-R1), and *TaK4* (using primers *Ta*K4-NdeI-F1 and *Ta*K4-BamHI-R1) were cloned into pMAL-C2 vector and expressed in *E. coli*. The sequences of the primers for the cloning and expression of the constructs encoding these peptides are provided in Supplementary Table 5.

*At*SEC and *At*SPY from Arabidopsis (Col-0) were used to test the *O*-GlcNAcylation enzyme activity on *Ta*GRP2. The full-length cDNA of *AtSEC*, which is the same as AT3G04240, was cloned into pSUMO vector using primers AtSEC-NdeI-F1 and AtSEC-BamHI-R1 (Supplementary Table 5). The full-length cDNA of *AtSPY*, which is the same as U62135, was cloned into pSUMO vector using primers AtSPY-EcoRI-F1 and AtSPY-BamHI-R1 (Supplementary Table 5). The proteins were expressed in *E. coli* (BL21), and the purified proteins were added in the *O*-GlcNAcylation reactions with *Ta*GRP2.

**Screening and identification of interacting proteins with TaOGT1**. A prey yeast-two hybrid (Y2H) library was constructed using winter wheat cultivar "2174"[41], which contains the same *vrn-A1b*, *PPD-D1b*, and *vrn-D3b* alleles as Billings. The *Ta*OGT1b protein was used as a bait to screen a Y2H library (Supplementary Method 5). Positive clones encoding proteins *Ta*K1 and *Ta*K4 were selected for further studies. The protein interaction experiments were performed to confirm the protein interactions in co-transformation of yeast cells and in BiFC experiments (Supplementary Method 5).

**Water-soluble carbohydrate (WSC) analysis**. WSC in leaves were extracted using the protocol of Ruuska et al.[69]. Extracts from 5 mg dried and ground material were used to measure sucrose, glucose and fructose content using a Sucrose/Fructose/D-Glucose Assay Kit (Megazyme, Bray, Ireland) according to the manufacturer's instructions. Contents of sucrose, glucose or fructose sugar were calculated according to a standard curve.

**Reporting summary**. Further information on research design is available in the Nature Research Reporting Summary linked to this article.

## Data availability
Data supporting the findings of this work are available within the paper and its Supplementary Information files. A reporting summary for this Article is available as a Supplementary Information file. The datasets and plant materials generated and analyzed during this study are available from the corresponding author upon reasonable request. Source data are provided with this paper.

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

## Acknowledgements

This project was supported by the Agriculture and Food Research Initiative Competitive Grants (2017-67007-25939) from the USDA National Institute of Food and Agriculture (NIFA), the NSF grant (OIA-1826820), and grants from the Oklahoma Center for Advanced Science and Technology (OCAST, AR17-020-03). M.F. received scholarships from the China Scholarship Council, Agricultural College of Nanjing Agricultural University, and the National Key Research and Development Program of China (2016YFD0100402) for her visit at the Oklahoma State University. H.J. was supported by the China 111 Project (B08025). This study was also supported by the Oklahoma Wheat Research Foundation and the Oklahoma Agricultural Experiment Station.

## Author contributions

M.F., F.M., H.J., G.L., C.P., R.N., and L.Y. performed the experiments and analyzed data. P.A. and B.C. phenotyped the population in the field and analyzed results. Z.M. advised M.F., and B.C. and L.Y. designed the experiments and directed the research. F.M. and L.Y. wrote the paper. All authors read and approved the paper.

## Competing interests

The authors declare no competing interests.
