## [Peer Review File · Nature Communications]

REVIEWER COMMENTS

Reviewer #1 (Remarks to the Author):

This manuscript of Fan et al. reports the significant roles of TaOGT1, a QTL gene, in regulating the flowering in winter wheat. The authors cloned the TaOGT1 gene from a QTL (QHd.osu-6A) on chromosome 6A regulating flowering based on different heading date by using two winter wheat cultivars "Duster" and "Billings". TaOGT1 encodes a previously uncharacterized O-linked N-acetylglucosamine (O-GlcNAc) transferase (OGT). Overexpression of TaOGT1b from Billings in Duster background promotes flowering with or without vernalization treatment. The further results showed that TaOGT1b overexpression reduced leaf glucose content, and increased the transcript levels of VRN1, VRN3 and PPD1. The authors also suggested that TaOGT1 had the O-GlcNAcylation enzyme activity, which can O-GlcNAcylate TaGRP2. Meanwhile, TaOGT1 activity was associated with sugar content and the transcript levels of flowering time genes. This information provides us a new idea to understand the mechanism of wheat vernalization and is of interest to scientists in the field of wheat breeding. But there are still some key problems and confusions that should be addressed.

1.As reported, CARG-box motif can be targeted by MADS-box transcription factors. I wonder why the authors think that TaVRN1 rather than other MADS-box TFs such as TaVRT2 binds to this CARG-box motif in the first intron of TaOGT1b. I also wonder the relationship between TaVRN1 and TaOGT1 (TaVRN1 directly regulates the transcription of TaOGT1 or TaOGT1 indirectly regulates the transcription of TaVRN1?). The authors should add more assays to confirm their related results.

2.When the authors used the transient promoter activity assays to assess TaOGT1a and TaOGT1b promoter and the the presence and absence of the 168-bp fragment activity, did the authors use the protoplasts of winter wheat (TaVRN1 is induced by prolonged cold exposure) or the protoplasts of spring wheat (TaVRN1 can be expressed without vernalization treatment)? To some extent, the protoplasts from winter wheat or spring wheat can be used to test whether TaVRN1 regulates the transcription of TaOGT1.

3.The positive control (such as AtSEC: has the O-GlcNAcylation enzyme activity) and the negative control (such as AtSPY: doesn't have the O-GlcNAcylation enzyme activity) need to be added to the TaOGT1 enzyme activity assay.

4.The authors should add the subcellular localization assay of the TaOGT1 to analyze the location of the TaOGT1 in cell in this manuscript.

5.The authors didn't identify TaGRP2 interacting with TaOGT1 through Y2H assay with a prey yeast-two hybrid (Y2H) library. Whereas, the further assay showed that TaOGT1 could O-GlcNAcylate TaGRP2. The authors should provide the results about the interaction between TaOGT1 and TaGRP2 in this manuscript.

6.As reported, the O-GlcNAcylated modification on TaGRP2 was induced by vernalization. If TaOGT1 can O-GlcNAcylate TaGRP2, when and how does the TaOGT1 function on TaGRP2 during vernalization in plant? And what's the transcriptional pattern of TaOGT1 in response to vernalization?

7.The authors showed that TaK1 can inhibit the function of TaOGT1 on TaGRP2's O-GlcNAcylation. How about the phenotype of TaK1 overexpression transgenic wheat? Is it opposite to the phenotype of TaOGT1 overexpression transgenic lines?

8.The authors should add the CBB staining result of the proteins used in the Fig.5a and Fig.5d, like the upper image in Fig.5b.

9.The showed images of PCR markers test in Fig S4 and the images of BiFC in Fig S16 are hard to read. The authors should provide the pictures with good quality.

10.In this manuscript, there are also lots of errors in the figures and the text. For example, there are several repeated figures in the manuscript (eg. Fig.1a is the same as Fig.S1a; Fig.1d is the partial same as Fig.S3); the authors also need to mark the names of different wheats in the fig1a and fig1b; the gene IDs of the QHd.osu-6A candidate gene are written wrong in the Fig.2a and Fig.2b, please correct them; the arrow is in the wrong position in Fig.5a; the "ranscript" should be replaced by "transcript" in Fig.2c and Fig.2d; "Fig.S17" and "Fig.S18" should replace "Fig.17" and "Fig.18" in the Supplementary information; the "aa 234-264" should replace "aa 234-64" in line 257. The authors should revise the manuscript accordingly and polish the language for easy reading.

Reviewer #2 (Remarks to the Author):

Fan et al., describe a novel gene TaOGT1 by map-based cloning which finely regulates flowering time in winter wheat cultivars. In particular, they establish TaOGT1–O-GlcNAc signal networks. These findings will help wheat researchers to understand better the mechanism of the minor differences in flowering time among winter wheat cultivars, and lay a foundation to breed wheat varieties adapted to climate change.

I have some comments on clarifications to text and figures for the authors to improve the manuscript.

1.Line179-180: The description "TaOGT1a in Duster was significantly higher than that of TaOGT1b in Billings" is not consistent with Fig.2c and 2d.

2.The candidate gene is TraesCS3D01G091300 in Fig.2a, 2b, but in the figure legend, it is TraesCS6A01G091300, why?

3.In Fig.3a, the endogenous control construct is created using the ubiquitin promoter, but in the figure legend, it is said the promoter is CaMV 35S. Why?

4.With regard to Fig.5,

1)There is no band pointed by the arrow in Fig.5a.

2)Line247: The description "TaGRP2 was significantly lower than in the control reaction without TaK1" is in contradiction with Fig.5b.

3)Why is one of fragment protein of TaOGT1 described as aa. 1-28 appeared in the most places, but in Fig.5d, it is shown as aa. 1-29?

4)Has TaK1 the same effect on the GlcNAcylation of TaOGT1 on TaGRP2 in transgenic and non-transgenic plants (Fig.5d)?

5.Line329: "No orthologous TPS1 has been identified in wheat", actually, you can find the orthologous genes of TPS1 in wheat when you blast TPS1(AT1G78580) in the database of Chinese Spring (IWGSCv1.0).

6.Other minor comments

1)Carefully check the format and other errors in the reference section, such as Line 507, 518, 528, 543, 578, 581 and 610.

2)Line743: a-i; Line746: g-l?

Reviewer #3 (Remarks to the Author):

The study by Fan et al describes the steps followed for cloning a flowering time QTL from wheat. The authors identify TaOGT a previously uncharacterized gene encoding an O-linked N-acetylglucosamine transferase that differentiates heading date between winter wheat cultivars. The authors developed a mapping population, detected a major QTL for flowering time and cloned a candidate gene. They discovered variation between the parents of the population, with an insertion of 168 bp in the first intron of TaOGT in the faster cultivar Billings. Overexpression of this gene accelerated flowering of transgenic Duster plants. Transgenic plants showed lower glucose and sucrose content. They also showed that TaOGT is able to transfer an O-GlcNAc group to plant protein TaGRP2 and E.coli proteins.

The work is original and the results will be of interest for many researchers working in plant science, and for practical application in plant breeding.

The authors followed a standard approach, i.e. development of a mapping population, searching for heading date QTL, fine mapping of the QTL, identification of three candidate genes, sequencing of those genes in the parents of the population, and validation by transgenic complementation. They also test the enzymatic activity of TaOGT1 and search for interacting proteins to it. The methodology is

well described, the results are quite clear and the conclusions are supported by the data provided.

The statistical analysis is appropriate, error bars are correctly defined in the figure legends. Please check and clarify the legend of Figure 6. "a-i. Comparison of gene expression in TaOGT1b overexpressed plants and non transgenic plants that were grown without vernalization (-) in the greenhouse (GH) or with vernalization (+) in a cold room (CR) for 1 week (1WK), 3 weeks (3WK), or 6 weeks (6WK)". Are '-' non-transgenic plants and '+' TaOGT1b overexpressing plants? Is the greenhouse treatment without vernalization and cold room with vernalization?

I have some minor comments or suggestions that should be revised to strengthen the study.

L114: 'Duster' and 'Billings' do not appear among the 18 winter wheat cultivars cited in the publication by Chen et al [17]. Have they changed the name? The genotype for the three developmental genes *vrn1*, *PPD1* and *vrn3* should be shown, at least in Supplementary material.

L179-181: Looking at Figures 2c and 2d, the result is that transcript levels of TaOGT1b in Billings are higher than those of TaOGT1a in Duster. Please check

L203: "... but the Billings TaOGT1a allele does not have this target site". Do you mean the Duster TaOGT1a allele?

L211: "This generated seven positive T0 plants". Six individual plants are mentioned in M&M, lines 370-371. Check

L274-277: Check the sentence. Transgenic TaOGT1b overexpressing lines non-vernalized showed higher expression levels of *vrn1*, *vrn3* and *PPD1* compared with 'the non-vernalized plants that were not vernalized either'. It must be non-transgenic plants not vernalized.

L284: "... of each flowering time gene in for all transgenic and non-transgenic plants..." They tested the transcript level of each flowering time gene 'in' or 'for' all transgenic and non-transgenic plants.

L378-379: "The samples were collected at 17:00". When were the lights turned on? Please clarify.

L610: It should be *Arabidopsis thaliana*, in italics.

L717: Check the parenthesis - "... protein (ng) and UDP) calculated from the UDP standard curve) in the samples..."

L743-745: Please clarify the legend of the figure. Are '-' non-transgenic plants and '+' TaOGT1b overexpressing plants? Is the greenhouse treatment without vernalization and cold room with vernalization?

All the outstanding comments were very helpful in revising the manuscript. We sincerely thank the three reviewers for their invaluable inputs. Below are detailed point-by-point responses to all comments made by the three reviewers.

Reviewer #1 (Remarks to the Author):

This manuscript of Fan et al. reports the significant roles of TaOGT1, a QTL gene, in regulating the flowering in winter wheat. The authors cloned the TaOGT1 gene from a QTL (QHd.osu-6A) on chromosome 6A regulating flowering based on different heading date by using two winter wheat cultivars “Duster” and “Billings”. TaOGT1 encodes a previously uncharacterized O-linked N-acetylglucosamine (O-GlcNAc) transferase (OGT). Overexpression of TaOGT1b from Billings in Duster background promotes flowering with or without vernalization treatment. The further results showed that TaOGT1b overexpression reduced leaf glucose content, and increased the transcript levels of VRN1, VRN3 and PPD1. The authors also suggested that TaOGT1 had the O-GlcNAcylation enzyme activity, which can O-GlcNAcylate TaGRP2. Meanwhile, TaOGT1 activity was associated with sugar content and the transcript levels of flowering time genes. This information provides us a new idea to understand the mechanism of wheat vernalization and is of interest to scientists in the field of wheat breeding. But there are still some key problems and confusions that should be addressed.

1. As reported, CARG-box motif can be targeted by MADS-box transcription factors. I wonder why the authors think that TaVRN1 rather than other MADS-box TFs such as TaVRT2 binds to this CARG-box motif in the first intron of TaOGT1b. I also wonder the relationship between TaVRN1 and TaOGT1 (TaVRN1 directly regulates the transcription of TaOGT1 or TaOGT1 indirectly regulates the transcription of TaVRN1?). The authors should add more assays to confirm their related results.

Response:

We agree that any MADS-box transcription factors including *TaVRN1* and *TaVRT2* could directly bind to the CARG element in intron one of *TaOGT1*. There are hundreds of MADS-box transcription factors in common wheat. Except for *TaVRN1*, *TaVRT2* has been studied. We chose *TaVRN1* in this study because this gene has been found to play important roles in developmental processes in our locally adapted winter wheat cultivars (Chen et al. 2009, Chen 2010, Li et al. 2013). Furthermore, *TaVRN1* promotes wheat development. *TaVRT2* was another MADS-box transcription factor that has been extensively studied, but the role of *TaVRT2* was controversial in two studies. *TaVRT2* was reported to be a repressor of wheat development (Kane et al., 2005; Kane et al., 2007), but *TaVRT2* was recently reported to be a promoter of wheat development (Xie et al., 2019).

In order to address the issue raised by the reviewer, we did additional experiments on *TaVRT2*. New results from the additional experiments included: 1). *TaVRT2* protein interacted with the 168-bp fragment of *TaOGT1* in an EMSA experiment (revised Fig. 3f). 2). *TaVRT2* was down-regulated by vernalization (revised Fig. 6j) but not significantly regulated in *TaOGT1b* over-expressed transgenic wheat, compared with the non-transgenic plants. The effects of vernalization on the transcript levels of *TaVRT2* in the same samples as tested for the other genes and their correlations were added in Table S1, S2 and S3. We added the new results and revised the Discussion in the revision.

Regarding the relationship between *TaVRN1* and *TaOGT1*, we revised the Discussion. On one hand, *TaVRN1* up-regulated the transcription of *TaOGT1* by the direct binding of *TaVRN1* protein with the *CArG* element in intron one of *TaOGT1*, which was indicated in normal wheat. On the other hand, the transcription of *TaVRN1* was upregulated by *TaOGT1*, which was shown in the *TaOGT1* over-expression transgenic wheat plants. Based on the gene transcription results, it is likely that there is a positive loop between *TaVRN1* and *TaOGT1*.

2. When the authors used the transient promoter activity assays to assess *TaOGT1a* and *TaOGT1b* promoter and the presence and absence of the 168-bp fragment activity, did the authors use the protoplasts of winter wheat (*TaVRN1* is induced by prolonged cold exposure) or the protoplasts of spring wheat (*TaVRN1* can be expressed without vernalization treatment)? To some extent, the protoplasts from winter wheat or spring wheat can be used to test whether *TaVRN1* regulates the transcription of *TaOGT1*.

Response:

In the original version, we used the protoplasts of spring wheat ‘Bobwhite’ without vernalization treatment in the transient promoter activity assays. The protoplasts can be efficiently extracted from seedlings of 10-15 cm in length that takes only 7-10 days to grow, but the winter wheat seedlings require a low temperature of a few weeks to vernalize. It could be difficult to test protoplasts using vernalized winter wheat cultivars.

We agree that the protoplasts from winter or spring wheat seedlings could be used to test whether the transcription of *TaOGT1b* with the 168-bp fragment is regulated by *TaVRN1*, *TaVRT2* or other MADS-box proteins. However, it requires long-term studies to do such an experiment involving various genetic backgrounds, different proteins even protein-protein complexes.

3. The positive control (such as *AtSEC*: has the O-GlcNAcylation enzyme activity) and the negative control (such as *AtSPY*: doesn’t have the O-GlcNAcylation enzyme activity) need to be added to the *TaOGT1* enzyme activity assay.

Response:

In the original version, we used a positive control provided in the reaction kit.

In order to address the issue raised by the reviewer, we requested *Arabidopsis* seed (*Col-0*), cloned cDNA copies and expressed proteins of *AtSEC*, which is the same as AT3G04240, and *AtSPY*, which is the same as U62135. We found that *TaGRP2* was GlcNAcylated by *AtSEC* but not by *AtSPY* (Fig. 5e, Fig. S19d in the revision). The results indicated that although *TaOGT1* and *AtSEC* have no similarity or domain conservation in protein sequences, they showed the similar GlcNAcylation activity on the same substrate as *TaGRP2*. We added the new results and revised the Discussion in the revision.

4. The authors should add the subcellular localization assay of the *TaOGT1* to analyze the location of the *TaOGT1* in cell in this manuscript.

Response:

The images of subcellular localization of the *TaOGT1* were provided in Fig. S16 (the first group). It is too hard to put the images in the main text.

5. The authors didn’t identify *TaGRP2* interacting with *TaOGT1* through Y2H assay with a prey yeast-two hybrid (Y2H) library. Whereas, the further assay showed that *TaOGT1* could O-

GlcNAcylate TaGRP2. The authors should provide the results about the interaction between TaOGT1 and TaGRP2 in this manuscript.

Response:

We did an additional experiment on *TaGRP2*, as suggested. The direct interaction between *TaOGT1b* and *TaGRP2* was observed in co-transformation of yeast cells (revised Fig. S15) and in a transient expression system with bimolecular fluorescence complementation (BiFC) in tobacco leaves (revised Fig. S16). We added the results in the revision.

6. As reported, the O-GlcNAcylated modification on TaGRP2 was induced by vernalization. If TaOGT1 can O-GlcNAcylate TaGRP2, when and how does the TaOGT1 function on TaGRP2 during vernalization in plant? And what's the transcriptional pattern of TaOGT1 in response to vernalization?

Response:

TaVRT2 was reported to be a repressor of wheat development (Kane et al., 2005; Kane et al., 2007), but *TaVRT2* was recently reported to be a promoter of wheat development (Xie et al., 2019). However, the result from this study showed that *TaVRT2* was down-regulated by low temperature (Fig. 6j).

The O-GlcNAcylated modification on *TaGRP2* was induced by vernalization. However, the result from this study showed that the transcriptional of *TaGRP2* was not significantly altered by vernalization (Fig. 6k). Interestingly, the transcript levels of *TaGRP2* were significantly increased in *TaOGT1b* over-expressed transgenic wheat, compared with the non-transgenic plants.

The transcriptional pattern of *TaOGT1* was provided in Fig. 6a. The downregulation of *TaOGT1* by low temperature was detectable in the 3 weeks-vernalized plants but not in the 6 weeks-vernalized plants.

Please note that the same cDNA samples were used to determine the transcript levels of TaOGT1, vrn1, VRN2, vrn3, PPD1, TaVRT2, TaGRP2, and actin. The RNA samples were collected at 17:00 before additional lights were turned on at 18:00 to make a long day condition, and the central part of the fresh leaves were analyzed as recommended. The leaf samples were from winter wheat cultivar 'Billings' that were tested in a temperature-controlled greenhouse. overexpressed transgenic plants (+) and non-transgenic plants (-). The plants were grown in the greenhouse (GH) without vernalization or were treated in a cold room (CR) with vernalization for 1 week (1 wk), 3 weeks (3 wk), or 6 weeks (6 wk). The transcripts of TaOGT1 in transgenic Duster plants included native TaOGT1a and transgenic TaOGT1b from Billings, whereas non-transgenic plants contained native TaOGT1a. The genotypes of the winter wheat cultivars Billings and Duster contain recessive vrn1 and vrn3 alleles and a photoperiod-insensitive allele, PPD1b. vrn1 indicates protein. It requires further studies in the future to establish direct and indirect relationships of the above-mentioned genes in genetic backgrounds and under different environments.

7. The authors showed that TaK1 can inhibit the function of TaOGT1 on TaGRP2's O-GlcNAcylation. How about the phenotype of TaK1 overexpression transgenic wheat? Is it opposite to the phenotype of TaOGT1 overexpression transgenic lines?

Response:

We agree that the function of *TaK1* should be validated using transgenic approach. However, we did not generate *TaK1* overexpression transgenic wheat in this study. The long-term experiments are out of the scope of the current manuscript.

8. The authors should add the CBB staining result of the proteins used in the Fig.5a and Fig.5d, like the upper image in Fig.5b.

Response:

We have the original CBB stained result for each of Western blot figures, but it was too hard to put so many CBB figures in the main text. We provided Fig. S19 of the CBB staining results in the revision for the four Western blot figures in Fig.5.

9. The showed images of PCR markers test in Fig. S4 and the images of BiFC in Fig. S16 are hard to read. The authors should provide the pictures with good quality.

The figures of PCR markers in Fig. S4 and the images of BiFC in Fig. S16 were improved.

10. In this manuscript, there are also lots of errors in the figures and the text. For example, there are several repeated figures in the manuscript (eg. Fig.1a is the same as Fig.S1a; Fig.1d is the partial same as Fig.S3); the authors also need to mark the names of different wheats in the fig1a and fig1b; the gene IDs of the *QHd.osu-6A* candidate gene are written wrong in the Fig.2a and Fig.2b, please correct them; the arrow is in the wrong position in Fig.5a; the “ranscript” should be replaced by “transcript” in Fig.2c and Fig.2d; “Fig.S17” and “Fig.S18” should replace “Fig.17” and “Fig.18” in the Supplementary information; the “aa 234-264” should replace “aa 234-64” in line 257. The authors should revise the manuscript accordingly and polish the language for easy reading.

Response:

In the main text,

Fig. 1a was replaced by a new figure for genotypes of *vrn1*, *PPD-D1*, and *vrn-D3*, as requested by reviewer 3.

The original Fig. S3 was deleted because the critical information was provided in Fig.1d.

The names of different wheats were marked under Fig.1a and Fig. 1b.

The gene IDs of the *QHd.osu-6A* candidate gene in the Fig. 2a and Fig. 2b were corrected.

The arrow Fig. 5a was placed at the right position.

The “ranscript” should be replaced by “transcript” in Fig. 2c and Fig. 2d;

The “aa 234-64” was replaced by “aa 234-264”.

In the Supplementary information,

“Fig.17” and “Fig.18” were replaced by “Fig. S17” and “Fig. S18” respectively.

Response: We have had the entire manuscript edited by professional editors. In addition, we have specifically addressed the instances noted by the other two reviewers.

Reviewer #2 (Remarks to the Author):

Fan et al., describe a novel gene *TaOGT1* by map-based cloning which finely regulates flowering time in winter wheat cultivars. In particular, they establish *TaOGT1*–O-GlcNAc signal

networks. These findings will help wheat researchers to understand better the mechanism of the minor differences in flowering time among winter wheat cultivars, and lay a foundation to breed wheat varieties adapted to climate change.

I have some comments on clarifications to text and figures for the authors to improve the manuscript.

1. Line179-180: The description “*TaOGT1a* in Duster was significantly higher than that of *TaOGT1b* in Billings” is not consistent with Fig.2c and 2d.

Response:

We really apologize for the mistake. We corrected the mistakes in the revision as follows.

We observed that the transcript level of *TaOGT1b* in Billings was significantly higher than that of *TaOGT1a* in Duster when each was grown at a variety of broad ambient temperatures either without (Fig. 2c) or with vernalization (Fig. 2d).

2. The candidate gene is TraesCS3D01G091300 in Fig.2a, 2b, but in the figure legend, it is TraesCS6A01G091300, why?

Response:

It was an error. TraesCS3D01G091300 in Fig.2a, 2b was changed to TraesCS6A01G091300.

3. In Fig.3a, the endogenous control construct is created using the ubiquitin promoter, but in the figure legend, it is said the promoter is CaMV 35S. Why?

Response:

It was an error. CaMV 35S was changed to *Ubiquitin*.

4. With regard to Fig.5,

1) There is no band pointed by the arrow in Fig.5a.

Response:

The arrow was moved at the right location.

2) Line247: The description “*TaGRP2* was significantly lower than in the control reaction without *TaK1*” is in contradiction with Fig. 5b.

Response:

In Fig. 5b, the GlcNAcylated *TaGRP2* protein (on the middle figure) in the reaction with *TaK1* showed significantly lower than that without *TaK1*.

The more *TaGRP2* proteins were GlcNAcylated, the stronger intensity of protein band representing the GlcNAcylated *TaGRP2* should be observed. The intensity of GlcNAcylated *TaGRP2* protein band with *TaK1* (lane 1) was lower than the intensity of GlcNAcylated *TaGRP2* protein band without *TaK1* (lane 2), indicating that the GlcNAcylation of *TaOGT1* on *TaGRP2* was repressed by *TaK1*.

We revised the description of the result.

The amount of GlcNAcylated *TaGRP2* proteins by *TaOGT1* in the GlcNAcylation reaction with *TaK1* was significantly lower than that in the control reaction without *TaK1* (Fig. 5b),

indicating that the GlcNAcylation of *TaOGT1* on *TaGRP2* was repressed by *TaK1*. However, *TaK4* showed no effect on the GlcNAcylation of *TaOGT1* on *TaGRP2* (Fig. 5b).

3) Why is one of fragment protein of *TaOGT1* described as aa. 1-28 appeared in the most places, but in Fig.5d, it is shown as aa. 1-29?

Response:

It was a mistake that *TaOGT1*-P1 was written as aa. 1-29 in Fig.5d. The mistake was corrected in the revision.

4) Has *TaK1* the same effect on the GlcNAcylation of *TaOGT1* on *TaGRP2* in transgenic and non-transgenic plants (Fig.5d)?

Response:

Fig. 5d was used to indicate that *TaOGT1* contains two active sites, when bacteria-expressed *TaOGT1*-P1 (aa 1–28) and *TaOGT1*-P4 (aa 234–264) were used as GlcNAcylation enzymes, *TaGRP2* was used as a substrate, and *TaK1* was added in the GlcNAcylation reactions in the study.

Fig. 5b was used to indicate that *TaOGT1* GlcNAcylated *TaGRP2* and *TaK1* repressed the GlcNAcylation, when the total proteins of transgenic wheat plants were used the source of GlcNAcylation enzymes. In Fig. 5b, the total proteins of from non-transgenic wheat plants were not used as a control, because the activity of the GlcNAcylation enzymes from the non-transgenic plants was not detectable, as shown in Fig. 5a.

5. Line329: “No orthologous *TPS1* has been identified in wheat”, actually, you can find the orthologous genes of *TPS1* in wheat when you blast *TPS1* (AT1G78580) in the database of Chinese Spring (IWGSCv1.0).

Response:

We apologize for the incorrect statement. Arabidopsis *TPS1* (AT1G78580) has orthologs in wheat in GenBank databases (e.g. ACI16353). The statement was revised in the Discussion as follows.

Based on sequence identity, orthologous wheat genes of Arabidopsis *TPS1* (AT1G78580) can be found (e.g. ACI16353 and KAF6991139), but none of them has been functionally characterized.

6. Other minor comments

1) Carefully check the format and other errors in the reference section, such as Line 507, 518, 528, 543, 578, 581 and 610.

The references were re-formatted, and all references in the revision were thoroughly checked.

2) Line743: a-i; Line746: g-l?

Response:

g-l was deleted. The legend was revised.

Reviewer #3 (Remarks to the Author):

The study by Fan et al describes the steps followed for cloning a flowering time QTL from wheat. The authors identify *TaOGT* a previously uncharacterized gene encoding an O-linked N-

acetylglucosamine transferase that differentiates heading date between winter wheat cultivars. The authors developed a mapping population, detected a major QTL for flowering time and cloned a candidate gene. They discovered variation between the parents of the population, with an insertion of 168 bp in the first intron of TaOGT in the faster cultivar Billings. Overexpression of this gene accelerated flowering of transgenic Duster plants. Transgenic plants showed lower glucose and sucrose content. They also showed that TaOGT is able to transfer an O-GlcNAc group to plant protein TaGRP2 and E. coli proteins.

The work is original and the results will be of interest for many researchers working in plant science, and for practical application in plant breeding.

The authors followed a standard approach, i.e. development of a mapping population, searching for heading date QTL, fine mapping of the QTL, identification of three candidate genes, sequencing of those genes in the parents of the population, and validation by transgenic complementation. They also test the enzymatic activity of TaOGT1 and search for interacting proteins to it. The methodology is well described, the results are quite clear and the conclusions are supported by the data provided.

The statistical analysis is appropriate, error bars are correctly defined in the figure legends. Please check and clarify the legend of Figure 6. “a–i. Comparison of gene expression in TaOGT1b overexpressed plants and non transgenic plants that were grown without vernalization (–) in the greenhouse (GH) or with vernalization (+) in a cold room (CR) for 1 week (1WK), 3 weeks (3WK), or 6 weeks (6WK)”. Are '-' non-transgenic plants and '+' TaOGT1b overexpressing plants? Is the greenhouse treatment without vernalization and cold room with vernalization?

Response:

We really apologize for the mistakes. We corrected the mistakes in the revision as follows.

a–i. Comparison of gene expression in *TaOGT1b* overexpressed transgenic plants (+) and non-transgenic plants (–). The plants were grown in the greenhouse (GH) without vernalization or were treated in a cold room (CR) with vernalization for 1 week (1 wk), 3 weeks (3 wk), or 6 weeks (6 wk).

I have some minor comments or suggestions that should be revised to strengthen the study.

L114: ‘Duster’ and ‘Billings’ do not appear among the 18 winter wheat cultivars cited in the publication by Chen et al [17]. Have they changed the name? The genotype for the three developmental genes *vrn1*, *PPD1* and *vrn3* should be shown, at least in Supplementary material.

Response:

We apologize for the citation mistake. We knew the genotypes of the two cultivars and used them in our breeding programs for many years, and we used Jagger and 2174 as controls as reported in Chen et al [17].

The genotype for the three developmental genes *vrn1*, *PPD1* and *vrn3* were provided in Fig. S1 in the revised supplementary material.

L179-181: Looking at Figures 2c and 2d, the result is that transcript levels of TaOGT1b in Billings are higher than those of TaOGT1a in Duster. Please check

Response:

We really apologize for the mistake. We corrected the mistakes in the revision as follows.

We observed that the transcript level of *TaOGT1b* in Billings was significantly higher than that of *TaOGT1a* in Duster when each was grown at a variety of broad ambient temperatures either without (Fig. 2c) or with vernalization (Fig. 2d).

L203: "... but the Billings TaOGT1a allele does not have this target site". Do you mean the Duster TaOGT1a allele?

Response:

We apologize for the mistake. It is Duster that has the *TaOGT1a* allele.

L211: "This generated seven positive T0 plants". Six individual plants are mentioned in M&M, lines 370-371. Check

Response:

It was corrected in M&M. We found 'seven' positive T₀ plants. Thanks.

L274-277: Check the sentence. Transgenic TaOGT1b overexpressing lines non-vernalized showed higher expression levels of *vrn1*, *vrn3* and PPD1 compared with 'the non-vernalized plants that were not vernalized either'. It must be non-transgenic plants not vernalized.

Response:

It was corrected as suggested. Thanks.

L284: "... of each flowering time gene in for all transgenic and non-transgenic plants..." They tested the transcript level of each flowering time gene 'in' or 'for' all transgenic and non-transgenic plants.

Response:

'for' was deleted.

L378-379: "The samples were collected at 17:00". When were the lights turned on? Please clarify.

Response:

The greenhouse has a natural light, but additional lights were provided to make a long day condition (16 h light/8 h darkness). The additional lights were turned on at 18:00.

L610: It should be *Arabidopsis thaliana*, in italics.

Response:

'*Arabidopsis thaliana*' herein was written in italics.

L717: Check the parenthesis – "... protein (ng) and UDP) calculated from the UDP standard curve) in the samples..."

Response:

The error was corrected as follows:

The *x* and *y* axes represent the amounts of protein (ng) and UDP (calculated from the UDP standard curve) in the samples.

L743-745: Please clarify the legend of the figure. Are '-' non-transgenic plants and '+' TaOGT1b overexpressing plants? Is the greenhouse treatment without vernalization and cold room with vernalization?

Response:

We really apologize for the mistakes. We corrected the mistakes in the revision as follows.

Comparison of gene expression in *TaOGT1b* overexpressed transgenic plants (+) and non-transgenic plants (-). The plants were grown in the greenhouse (GH) without vernalization or were treated in a cold room (CR) with vernalization for 1 week (1 wk), 3 weeks (3 wk), or 6 weeks (6 wk).

REVIEWERS' COMMENTS

Reviewer #1 (Remarks to the Author):

The revised manuscript has basically addressed my main concerns. But there are still some errors that should be corrected.

1. The description is wrong in lines 273 and 274, in Arabidopsis, AtSPY is reported to have O-Fucosyltransferase enzyme activity rather than O-GlcNAcylation enzyme activity; Please read the references carefully, correct the related descriptions about AtSPY and update the references in this manuscript.

(Zentella et al. (2017). The Arabidopsis O-fucosyltransferase SPINDLY activates nuclear growth repressor DELLA. Nat. Chem. Biol 13:479–485

Wang et al.(2019) Nuclear Localized O-Fucosyltransferase SPY Facilitates PRR5 Proteolysis to Fine-Tune the Pace of Arabidopsis Circadian Clock, Mol Plant.13(3): 446–458.)

2. The description is wrong in lines 319 and 320, the reference 40 showed that the transcription of TaGRP2 is not regulated by vernalization, but the O-GlcNAcylation modification on TaGRP2 is induced by vernalization, and TaGRP2's subcellular localization is changed by vernalization. Please read the references carefully and correct the related descriptions about TaGRP2 in this text.

3. The "O" in O-GlcNAc" should be italic, the authors should correct it through the whole manuscript;

4. TaOGT1b in lines 230, 232, 250, 285, 289, 291, 402, 403, 404 should be italic;

5. At first mention (line 246) please define "GR-RBP" in line 246;

6. There should be blank among 0.2M in line 420.

Reviewer #2 (Remarks to the Author):

The authors have addressed my comments and have modified the manuscript except for aa1-29 in the Fig.5d.

Reviewer #3 (Remarks to the Author):

The authors have addressed all the points raised during the previous review, improving the manuscript with new results and clarifying some other issues.

There are two minor typos that can still be fixed in the current manuscript, as follows.

L757, legend Figure 2: "... insertion in intron 1 of the Billing allele is indicated... ". Is it the Billings allele?

L797, legend Figure 3: "... bp fragment, were used a promoter of the GUS reporter gene". Is it used as promoter?

[Editor: The newly added Reviewer #4 states in Remark to Editor section that the related glycobiology assays are technically sound and the conclusions are supported by the available data.]

REVIEWERS' COMMENTS

Reviewer #1 (Remarks to the Author):

The revised manuscript has basically addressed my main concerns. But there are still some errors that should be corrected.

1. The description is wrong in lines 273 and 274, in arabidopsis, AtSPY is reported to have O-Fucosyltransferase enzyme activity rather than O-GlcNAcylation enzyme activity; Please read the references carefully, correct the related descriptions about AtSPY and update the references in this manuscript (Zentella et al. (2017). The Arabidopsis O-fucosyltransferase SPINDLY activates nuclear growth repressor DELLA. Nat. Chem. Biol 13:479–485 Wang et al. (2019) Nuclear Localized O-Fucosyltransferase SPY Facilitates PRR5 Proteolysis to Fine-Tune the Pace of Arabidopsis Circadian Clock, Mol Plant.13(3): 446–458.)

We updated information about *AtSPY*, and changed it as follows.

The Arabidopsis SPINDLY (*AtSPY*) was reported to be an OGT protein [31] but it was recently believed to be an *O*-fucosyltransferase [32, 33].

We kept the old information for *AtSPY* [31], so readers can ready understand why *AtSPY* was selected as a control in this study. We added two new references as suggested [32, 33], and we also changed relevant sentences with *AtSPY* throughout the revised text.

2. The description is wrong in lines 319 and 320, the reference 40 showed that the transcription of TaGRP2 is not regulated by vernalization, but the O-GlcNAcylation modification on TaGRP2 is induced by vernalization, and TaGRP2's subcellular localization is changed by vernalization. Please read the references carefully and correct the related descriptions about TaGRP2 in this text.

It has been changed as advised. Thanks.

The O-GlcNAcylation modification on *TaGRP2* is induced by vernalization (46).

3. The "O" in O-GlcNAc" should be italic, the authors should correct it through the whole manuscript;

It was changed through the whole manuscript.

4. TaOGT1b in lines 230, 232, 250, 285, 289, 291, 402, 403, 404 should be italic;

It was changed in these cases and checked through the text.

5. At first mention (line 246) please define "GR-RBP" in line 246;

It was defined as be a glycine-rich RNA-binding protein (GR-RBP or GRP).

6. There should be blank among 0.2M in line 420.

It was corrected.

Reviewer #2 (Remarks to the Author):

The authors have addressed my comments and have modified the manuscript except for aa1-29 in the Fig.5d.

'aa1-29' was changed to 'aa1-28'.

Reviewer #3 (Remarks to the Author):

The authors have addressed all the points raised during the previous review, improving the manuscript with new results and clarifying some other issues.

There are two minor typos that can still be fixed in the current manuscript, as follows.

L757, legend Figure 2: "... insertion in intron 1 of the Billing allele is indicated... ". Is it the Billings allele?

'Billing' was changed to 'Billings'.

L797, legend Figure 3: "... bp fragment, were used a promoter of the GUS reporter gene". Is it used as promoter?

'as' was added before a promoter.

[Editor: The newly added Reviewer #4 states in Remark to Editor section that the related glycobiology assays are technically sound and the conclusions are supported by the available data.]